# Comprehensive Review on the Impact of Chemical Composition, Plasma Treatment, and Vacuum Ultraviolet (VUV) Irradiation on the Electrical Properties of Organosilicate Films

**DOI:** 10.3390/polym16152230

**Published:** 2024-08-05

**Authors:** Mikhail R. Baklanov, Andrei A. Gismatulin, Sergej Naumov, Timofey V. Perevalov, Vladimir A. Gritsenko, Alexey S. Vishnevskiy, Tatyana V. Rakhimova, Konstantin A. Vorotilov

**Affiliations:** 1Research and Educational Center “Technological Center”, MIREA—Russian Technological University (RTU MIREA), 119454 Moscow, Russia; vorotilov@live.ru; 2European Centre for Knowledge and Technology Transfer (EUROTEX), 1040 Brussels, Belgium; 3Rzhanov Institute of Semiconductor Physics SB RAS, 13 Lavrentiev Ave., 630090 Novosibirsk, Russia; aagismatulin@isp.nsc.ru (A.A.G.); timson@isp.nsc.ru (T.V.P.); grits@isp.nsc.ru (V.A.G.); 4Leibniz Institute of Surface Engineering (IOM), 04318 Leipzig, Germany; sergej.naumov@iom-leipzig.de; 5Automation and Computer Engineering Department, Novosibirsk State Technical University, 20 Marks Ave., 630073 Novosibirsk, Russia; 6Skobeltsyn Institute of Nuclear Physics, Lomonosov Moscow State University (SINP MSU), 119991 Moscow, Russia; trakhimova@mics.msu.su

**Keywords:** organosilicate glass (OSG), low-*k* dielectrics, thin film, electrical properties degradation, charge transport, trap energy, chemical composition, bonding configuration, density functional theory (DFT), photoluminescence

## Abstract

Organosilicate glass (OSG) films are a critical component in modern electronic devices, with their electrical properties playing a crucial role in device performance. This comprehensive review systematically examines the influence of chemical composition, vacuum ultraviolet (VUV) irradiation, and plasma treatment on the electrical properties of these films. Through an extensive survey of literature and experimental findings, we elucidate the intricate interplay between these factors and the resulting alterations in electrical conductivity, dielectric constant, and breakdown strength of OSG films. Key focus areas include the impact of diverse organic moieties incorporated into the silica matrix, the effects of VUV irradiation on film properties, and the modifications induced by various plasma treatment techniques. Furthermore, the underlying mechanisms governing these phenomena are discussed, shedding light on the complex molecular interactions and structural rearrangements occurring within OSG films under different environmental conditions. It is shown that phonon-assisted electron tunneling between adjacent neutral traps provides a more accurate description of charge transport in OSG low-*k* materials compared to the previously reported Fowler–Nordheim mechanism. Additionally, the quality of low-*k* materials significantly influences the behavior of leakage currents. Materials retaining residual porogens or adsorbed water on pore walls show electrical conductivity directly correlated with pore surface area and porosity. Conversely, porogen-free materials, developed by Urbanowicz, exhibit leakage currents that are independent of porosity. This underscores the critical importance of considering internal defects such as oxygen-deficient centers (ODC) or similar entities in understanding the electrical properties of these materials.

## 1. Introduction

Organosilicate glasses (OSGs) constitute a family of organic–inorganic hybrid materials characterized by a silica-like backbone structure. By incorporating diverse organic moieties into a silica matrix and employing various deposition techniques, the properties of these materials can be finely tuned. The potential practical applications of hybrid materials span a broad spectrum, encompassing areas such as adsorption, catalysis, microelectronics, and bioengineering. Since the late 1990s, dense and porous variants of these materials have been widely used in advanced microelectronics, stimulating further research and development efforts focused on depositing thin films and engineering their properties [1,2,3].

At present, organosilica films are deposited using chemical vapor deposition (CVD), plasma-enhanced CVD (PECVD), and hot-filament CVD (HFCVD), as well as chemical solution deposition (CSD), which involves sol–gel chemistry with deposition methods such as spin-coating and dip-coating. In the first method, the films incorporate methyl terminal groups on their pore wall surfaces. However, the processes governing film formation lack precise control due to the interaction of numerous radicals and intermediates formed within the CVD and plasma reactor. Sol–gel-based methods offer greater control over deposited material properties because the selected precursors exhibit relative stability and do not undergo deep fragmentation during processing. The reactions primarily proceed through the hydrolysis of terminal alkoxy groups followed by condensation reactions to form the molecular skeleton. Furthermore, the application of some special techniques, such as evaporation-induced self-assembly (EISA), can achieve a more uniform pore size distribution and a defined spatial arrangement [4]. In addition, some of the oxygen bridging groups (≡Si–O–Si≡) in SiO_2_ matrix can be substituted with carbon bridges (≡Si–C*_x_*H*_y_*–Si≡) to enhance the elasticity of the matrix material because of the higher bending rigidity of these groups [5]. These materials are termed periodic mesoporous organosilicas (PMOs).

Numerous properties of these materials have undergone extensive study and documentation over the past two decades, with a wealth of information available in various review papers and monographs [6,7,8,9]. However, unraveling the complexities of their electrical characteristics poses a significant challenge, as accurate measurement and conclusive interpretation heavily rely on factors such as deposition and curing processes, test structure preparation, and instrumentation quality. In an effort to address these challenges, we conducted a comprehensive analysis of both existing and newly acquired data. It is noteworthy that numerous recent publications have focused on the reliability and properties of integrated low-*k* dielectrics [10]. In our study, however, we have limited our analysis to non-integrated OSG dielectrics. This approach aims to elucidate the origin of electrically active defects prior to their modification through the integration process. To enhance clarity and coherence, the paper is structured into three distinct sections.

The first section of this paper provides a general introduction. The second section focuses on fabricating porous organosilica films, discussing deposition strategies from both gas phase (plasma) and liquid phase (sol–gel technology and spin-on deposition). It also covers methods for generating porosity, types of sacrificial porogen/templates, and UV curing of the deposited films. Furthermore, the section explores the differences between OSGs fabricated using PECVD and sol–gel methods, explaining the functions of methyl terminal and carbon bridging groups.

The third section introduces the modification of materials resulting from the utilization of various plasma systems and processes employed for micropatterning, which are crucial for practical applications. This is followed by an analysis of vacuum ultraviolet (VUV)-induced modifications, detailing the energy and mechanisms involved in bond breakage within OSG materials. Furthermore, it delves into a discussion of different types of electrically active defects.

The fourth section includes an analysis of data related to electrical properties. It is demonstrated that the matrix of OSG films containing methyl terminal groups exhibits a breakdown field similar to that of amorphous SiO_2_. This similarity arises from the methyl groups being exclusively situated on the pore wall surface. Moreover, the incorporation of carbon-based bridging groups, instead of oxygen bridges, into the silica matrix leads to a reduction in the breakdown field. This reduction is more pronounced when aromatic groups are used as bridges compared to alkyl chains. The breakdown field decreases as porosity increases. In most cases of porogen-based low-*k* materials (with a lower relative dielectric constant *k* than that of SiO_2_, where *k* ≈ 3.9), this reduction is attributed to a higher concentration of porogen residue in highly porous films. The porogen residue contributes to the formation of a valence band tail, and this effect intensifies after plasma treatment and/or ion bombardment.

The mechanism of leakage current was investigated using spin-on and PECVD deposited films. Previous research generally attributes leakage current to the Poole–Frenkel mechanism under low electric fields and the Fowler–Nordheim mechanism under high electric fields. However, our study suggests that the mechanism involving phonon-assisted electron tunneling between adjacent neutral traps (Nasyrov–Gritsenko model) provides a more accurate description of charge transport in OSG low-*k* materials. The implications of this concept are discussed in greater detail.

## 2. Fabricating Organosilica Films

Fabrication of organosilica films involves the synthesis of hybrid materials that combine organic and inorganic components using various deposition methods, such as the following:

### 2.1. Chemical Vapor Deposition (CVD, PECVD, and HFCVD)

Chemical vapor deposition (CVD) is a process in which precursors are introduced into the gas phase and then transferred to a vacuum reaction chamber with a heated substrate [11,12]. The heat provides the thermal energy required for the reaction of the vaporized precursors to form the desired layer on the substrate.

Plasma-enhanced CVD (PECVD) involves the utilization of plasma to activate the monomer precursor [13]. The vaporized precursor molecules within the reaction chamber are bombarded by unbound electrons. This process generates more unbound electrons, ions, radicals, atoms, and molecules, leading to the activation and formation of reaction intermediates. These reaction intermediates polymerize in the gas phase and/or on the sample surface, ultimately resulting in the formation of a thin film. One primary benefit of PECVD is its capability to enable deposition at relatively low temperatures while ensuring uniformity across extensive surface areas. In PECVD, a number of factors are essential for creating dielectric films of superior quality. The primary factors to consider are the temperature of the substrate, the pressure, the power of the radio frequency (RF), and the ratios of the reactant gases’ flow. PECVD is currently the preferred method in the microelectronic industry due to its ability to seamlessly integrate into the device manufacturing process.

One alternative method used for the organosilica film deposition is hot-filament chemical vapor deposition (HFCVD). HFCVD-deposited materials do not suffer from the UV irradiation and ion bombardment associated with plasma exposure. In addition, HFCVD allows for more control over precursor fragmentation pathways than PECVD. Thermal activation is limited to the gas phase, and independent control of the substrate temperature can be exercised [14].

### 2.2. Spin-Coating Deposition

Spin-coating is using liquid precursors for deposition of layers such as resist, spin-on-glass (SOG), spin-on-diffusion (SOD), etc. This technique is suitable for ensuring uniformity and minimizing defects in film manufacturing. The main drawback is the high consumption of a rather expensive precursor composition (3–5 mL per application).

Spin-on deposition includes the following steps: deposition, spin-up, spin-off, and evaporation [15,16]. During the first two steps, the liquid is dispensed onto the wafer and spreads out due to centrifugal forces. The spin speed at these stages is typically low (hundreds of rpm). During the spin-off stage, the spinning speed increases (up to thousands of rpm), and the liquid flows under centrifugal force. Over time, the rate of decrease in film thickness due to the convective flow slows down because the convective flow is proportional to the cube of the film thickness. At the final stage, the viscosity increases sharply, and the convective outflow stops.

A very important application of the spin-coating process is related to local planarization or gap filling. The film profile tends to smooth or planarize the substrate features [17,18]. The planarization coefficient decreases as the drying rate decreases, along with a reduction in spin speed and an increase in aspect ratio. The deposition step is followed by heating or a “soft bake” step to remove solvent and initiate cross-linking of the film at temperatures typically ~150–200 °C. Finally, sintering at temperatures ranging from 400 to 430 °C, also known as a “hard bake” or “curing”, is necessary to initiate the final cross-linking of the polymer chains, resulting in a mechanically stable film.

Although PECVD is the current deposition technique due to its better compatibility with traditional semiconductor manufacturing, spin-coating shows promise as an alternative. It is more flexible in terms of use, with many precursors based on sol–gel chemistry reactions that are not accessible in gas-phase deposition. Liquid-phase deposition is highly suitable for preparing advanced ultra-low-*k* (*k* < 2.0) materials. Finally, planarization is a highly anticipated feature for use in the back-end-of-line (BEOL) subtractive process [19].

### 2.3. Matrix and Precursors

Organosilica materials developed for microelectronics applications have a silica-like matrix (Figure 1a). However, some of the bridging oxygen atoms are replaced by terminal methyl (or alkyl) groups, rendering the material hydrophobic (Figure 1b). This hydrophobization is crucial because water molecules possess a very high dielectric constant (~80), and even a small amount of adsorbed moisture can drastically increase the dielectric constant of the material. This issue is exacerbated if the material is porous, as the high surface area can adsorb a significant amount of moisture. Consequently, the precursors used for the deposition of organosilica films typically contain at least one methyl group directly bonded to silicon.

### 2.4. Porous Materials

Reduction in the dielectric constant is achieved by incorporating chemical bonds with low polarizability, such as Si–C, C–H, C–C, Si–H, Si–F, and C–F. More aggressive reduction is needed in the low-*k* materials with low density, and it can be achieved through two approaches.

The first method involves increasing the free volume by incorporating space-occupying groups such as methyl, ethyl, and phenyl, which rearrange the material structure and also decrease the dielectric constant. This porosity, known as constitutive porosity, typically results in micro-sized pores (*R* ≤ 1 nm) but is limited to about 10–15% [6]. In chemical vapor deposition (CVD) processes, porosity can be generated by optimizing the ratio of gas phase to surface reactions. If the reactant intermediates agglomerate in the gas phase, conditions can be achieved where the deposited films have open porosity up to ~30%.

The second method involves adding sacrificial substances during the manufacturing of the low-*k* film [20]. These substances are co-deposited with the matrix material and can be thermally removed. External impacts such as infrared (IR) or ultraviolet (UV) light and electron beams increase the rate and efficiency of porogen removal, thereby generating porosity. This approach, known as subtractive porosity, can exceed 50%, with the pore size depending on the type, molecular weight, and amount of porogen used (Table 1) [21]. Introducing porosity reduces the dielectric constant because air has a *k* value of ~1.0. However, this also weakens the mechanical stability of the material. Furthermore, porosity decreases the plasma and chemical stability of the dielectric material, leading to the adsorption of impurities. This degradation results in an increased dielectric constant (*k*), higher leakage current, and a reduced breakdown field.

The detailed mechanism of plasma-chemical reactions occurring during PECVD of organosilica films, followed by thermal (or UV-assisted) treatment, is quite complex and depends on the type of precursor molecules, plasma conditions, and temperature. The deposited films typically exhibit Si–O–Si bonds (observed at 1000–1100 cm^−1^ in FTIR spectra) and SiCH_3_ bonds (near 1275 cm^−1^). The presence of porogen molecules is mainly indicated by absorption in the range of 2800–3000 cm^−1^. As-deposited films are also hydrophilic and contain a significant amount of silanols and adsorbed moisture, which is detected in the range of 3000–4000 cm^−1^, with a SiOH peak at 3700 cm^−1^. Thermal curing removes porogen, indicated by the reduction in hydrocarbon absorption in the range of 2800–3000 cm^−1^ and also indicated by changes in the indices of refraction, suggesting the generation of porosity. This structural rearrangement improves the mechanical properties by densifying the skeleton [22] and pushing CH_3_ groups onto the pore wall surface. Non-optimized curing can remove some SiCH_3_ groups, leading to the formation of dangling bonds that are then saturated by hydrogen, forming hydrophilic SiH groups [23]. Detachment of CH_3_ groups is one of the key issues for degradation of OSG glasses, and it will be discussed in more detail in the section related to VUV and plasma modification.

The key chemical reaction occurring during the curing is condensation of silanols
≡Si–OH + HO–Si≡ → ≡Si–O–Si≡ + H_2_O (condensation)(1)
that provides the skeleton densification and hydrophobization. According to the findings reported by Gourhant et al. [24], the condensation can also involve other groups:≡Si–O–CH_2_–CH_3_ + SiCH_3_ → ≡Si–O• + ≡Si• + CH_3_–CH_2_–CH_3_ → ≡Si–O–Si≡ + C*_x_*H*_y_*(2)

Finally, the OSG film has to become sufficiently hydrophobic with a densified skeleton providing relatively good mechanical properties.

In the liquid phase, pores are incorporated through the addition of thermally or chemically labile precursors, pore generators, or surfactants [1,2]. These pore generators and surfactants can be removed after deposition through thermal treatment, preferably assisted by UV light, effectively leaving behind a porous structure. The excess volatile solvent keeps the surfactant concentration below the critical micelle concentration (CMC). Upon solvent evaporation following film deposition, the surfactant concentration exceeds the CMC, initiating the self-assembly of silane precursor molecules around micelles. Afterward, a thermal curing step is conducted to facilitate further condensation of the silica matrix (pore walls). The surfactant (Table 2) can be removed not only by heat treatment under an inert atmosphere via hydrothermal or evaporation-induced self-assembly (EISA) procedures but also through extraction. Mesostructured ordering, such as hexagonal or cubic structures, can be achieved through this process, which was first reported by Brinker’s group using the dip-coating method [25,26,27].

Many researchers have noted that ionic surfactants containing halogens, such as cetyltrimethylammonium chloride (CTAC) and cetyltrimethylammonium bromide (CTAB), result in minimal pore size. For instance, a CTAC-templated film with a pore size of 1.7 nm is more hydrophilic compared to films templated by Brij^®^ L4 (also known as Brij^®^ 30), Brij^®^ C2 (also known as Brij^®^ 52), Brij^®^ C10 (also known as Brij^®^ 56), and Brij^®^ S10 (also known as Brij^®^ 76), which have larger pore sizes [28]. CTAB leads to the best low-*k* properties and Young’s modulus compared to Brij^®^ 76 and Pluronic^®^ F127 [29]. CTAB, with its lower molecular weight, enables the formation of pores with a radius of up to 1.1 nm while maintaining high porosity (*V_open_* = 49%) compared to Brij^®^-type surfactants. CTAB offers advantages such as a lower CMC, improved solubility, and a higher decomposition temperature compared to nonionic surfactants [30]. Moreover, ionic surfactants provide long-range order [31], thicker pore walls, and improved chemical resistance [29]. Nevertheless, Ting et al. [32] demonstrate that films produced with ionic surfactants exhibit excess negative charges, leading to ultrahigh leakage current. Consequently, the flatband voltage of nonionic template films shifts to a more negative value with increased porosity, suggesting an increase in positive charges within porous films. The leakage current density rises exponentially with porosity in nonionic template films [32].

In summarizing the utility of porous low-*k* materials, it is important to note that achieving a *k* value ≤ 2 (ultra-low-*k*) requires a high level of porosity exceeding 50%. Such porous materials do not possess the necessary mechanical stability required by interconnects. Consequently, alternative materials must be identified to meet these criteria. The potentially promising materials might be periodic mesoporous organosilicas (PMOs). However, despite their good mechanical properties, the carbon-bridged PMO materials do not exhibit sufficient hydrophobicity. They adsorb moisture, which significantly degrades their *k* value and increases leakage current [33]. For this reason, it is necessary to introduce methyl terminal groups to achieve a sufficient degree of hydrophobicity, but in the right quantity to minimize deterioration in the mechanical properties [34]. This makes it difficult to control the performance of these materials due to the challenge of effectively adjusting the ratio of bridging to terminal carbon groups to maintain both their favorable mechanical properties and the necessary hydrophobic nature. 

To effectively develop on-chip interconnections of ULSI devices (ILDs), which is the most demanding area for utilizing OSG thin films, it is crucial to have a comprehensive understanding of their fundamental characteristics [19,35].

As already mentioned, in the case of spin-on OSGs, the synthesis occurs by means of a sol–gel reaction [36]. As a starting precursor, alkoxy organosilanes or halogenated organosilanes can be used. By introducing a solvent, water, and a catalyst (acid or base), the precursor initiates the process of hydrolysis (Equation (3)) and subsequently undergoes condensation (Equation (1)). The final material can be produced using either the same starting precursors or different ones.
≡Si−OCH_3_ + H_2_O → ≡Si−OH + CH_3_−OH (hydrolysis)(3)

If one of the precursors contains a carbon bridge (like BTMSE), the bridge is incorporated into the wall structure. If the EISA process is used, the final PMO films might have ordered porosity, as shown in Figure 1c, with carbon bridges in their matrix and the methyl groups located on the pore wall surface.

### 2.5. Basic Characterization

#### 2.5.1. Chemical Composition and Bonds Configuration

The basic chemical composition of OSG films is analyzed using *Fourier-transform infrared (FTIR) spectroscopy*. These spectra clearly show the presence, absence, and behavior of bond content, which change their dipole moment upon the absorption of IR light. These bonds include polar bonds, such as Si–O–Si, Si–OH, Si–CH_3_, C–H_3_, C–H_2_, H–O–H, C–C, etc.

Figure 2 shows typical FTIR spectra of four different OSG films. The first film (Sample 1) is a methyl-terminated methylsilsesquioxane (MSSQ) film. Normally, PECVD OSG films exhibit similar spectra. Samples 2–4 are pure PMO films deposited without methyl terminal groups (with 100 mol% concentrations of methylene, ethylene, and benzene bridging precursors for Samples 2, 3, and 4, respectively). The most intense peaks in the FTIR spectra of all films are associated with the siloxane matrix (Si–O–Si stretching vibrations at 1300–1000 cm^−1^). The methyl-terminated film (Sample 1) has an intense Si–CH_3_ peak at ~1275 cm^−1^ [37,38]. Peaks associated with bridging groups are less pronounced and are mainly located in the region of 1700–1300 cm^−1^. These peaks appear weak in FTIR spectra due to their low concentration and significantly lower absorption coefficient compared to Si–O–Si bonds. However, the spectrum of Sample 4 exhibits the largest number of distinctive absorption bands because of the presence of bridging 1,4-phenylene (p-disubstituted) rings in its structure, particularly at ~1600 and ~1510 cm^−1^ [39,40,41]. The very weak band characteristic of C–H bonds in methylene bridges in Sample 2 is located at ~1360 cm^−1^, a position notably distinct from the peak of C–H bonds in ethylene bridges in Sample 4 (~1415 cm^−1^) [42]. Also, the C–C bonds of ethylene bridges absorb at ~720 cm^−1^ [43,44]. Some differences are also visible in the region of 3000–2850 cm^−1^ absorption of CH_3_ and CH_2_. The most significant observation is the increased CH_2_ intensity at 2950–2850 cm^−1^ in ethylene-bridged films. Silanol groups (Si–OH) absorb at ~950 and 3800–3200 cm^−1^ and are mainly observed in Sample 4. In turn, surface silanols serve as adsorption centers for water molecules, which are evident as a broad band at 3600–3200 cm^−1^. When a significant amount of water molecules is adsorbed by the film, the H–O–H peak at ~1630 cm^−1^ becomes clearly visible [45].

The bonding configuration can be evaluated through *X-ray photoelectron spectroscopy (XPS)* studies [46,47]. The XPS investigation focuses on analyzing the core energy levels of Si 2p, C 1s, and O 1s. For instance, an important feature can be seen in Si 2p XPS spectra (Figure 3). One can observe that the integral Si 2p peak position of the OSG film is broader and shifted to the low-energy region compared to SiO_2_, resembling suboxide or SiO*_x_*, where *x* < 2 [37]. This peak can be interpreted as containing additional compounds with valence in comparison with SiO_2_ (Si^4+^).

Deconvolution of the Si 2p peak is also conducted using four Gaussians, assigning them to the so-called M, D, T, and Q groups, which correlate with OSi(–C)_3_ (101.9 ± 0.1 eV), O_2_Si(–C)_2_ (102.9 ± 0.1 eV), O_3_Si–C (103.7 ± 0.1 eV), and SiO_4_ (104.4 ± 0.1 eV) configurations, respectively [47,48,49].

#### 2.5.2. Porosity and Pore Structure

Porosity and pore structure are critical properties of OSG low-*k* films because they define dielectric constant and compatibility with integration processes. Traditional porosimetry techniques have limitations in thin films because of their small total pore volume. Recent methods such as ellipsometric porosimetry (EP), X-ray porosimetry, neutron scattering contrast matching, and solvent adsorption are utilized to determine pore sizes in thin, porous films. Non-intrusive methods such as small-angle neutron and X-ray scattering spectroscopy, specular X-ray reflectivity (XRR), and positron annihilation lifetime spectroscopy (PALS) are also employed. Grazing-incidence small-angle X-ray scattering (GISAXS) spectroscopy is useful for evaluating the 3D mesostructure, pore arrangement, spacing, and structural order. Different techniques offer unique insights into pore structure and demonstrate good consistency in determining pore size and porosity. Radiation scattering techniques provide information about pore size, while X-ray reflectivity (XRR) is used to measure film density. Pore connectivity is an important characteristic of low-*k* films, affecting the diffusion of technological chemistries. Some low-*k* films have closed pores that restrict positronium diffusion but allow molecules of adsorbates such as toluene, isopropyl alcohol, etc., to pass through. PALS is advantageous for analyzing extremely small isolated (closed) pores and can evaluate pore sealing efficiency and interconnectivity. EP is simple and versatile, providing quantitative information on pore morphology, size distribution, surface area, mechanical properties, and overall porosity. EP is a simple and versatile technique that provides quantitative information on pore morphology, size distribution, surface area, mechanical properties, and porosity [50].

## 3. Modification of OSG by Plasma and VUV Radiation

Modification of OSG dielectrics is inevitable during their manufacture and integration. As previously mentioned, UV-assisted thermal curing is widely used to remove porogen and cross-link the low-*k* film matrix. Additionally, producing the final integrated products requires patterning the films according to specified tasks. This patterning typically involves plasma processing, which can modify low-*k* films through the action of energetic ions, VUV photons, and chemically active radicals.

Patterning is a critical stage in the fabrication of integrated circuits, demanding precision at the nanometer level to cater to the needs of advanced technology nodes in ULSI. With ULSI elements shrinking to sizes in the order of tens of nanometers, the intricacy of the process becomes even more apparent. While lithography establishes the initial target size, the real challenge lies in faithfully replicating that pattern onto the functional layer with utmost accuracy. It is a sophisticated process that underscores the remarkable advancements in semiconductor technology.

The emergence of the Damascene process in the late 1990s indeed marked a significant shift in interconnect technology within the semiconductor industry. This transition was primarily driven by the replacement of aluminum (Al) with copper (Cu) as the preferred (low resistivity) metal for interconnects. Unlike aluminum, copper cannot be effectively etched using plasma, necessitating a move away from subtractive technology based on aluminum. In the Damascene process, the sequence of steps is altered to accommodate the properties of copper. Initially, the dielectric layer is deposited and patterned, and then diffusion barriers, a special liner, and seeds are deposited before the metal deposition takes place. This sequence ensures that the dielectric layer acts as a template for the subsequent metal filling step. The filling of the patterned dielectric with copper is achieved through superfilling techniques. These techniques are designed to deposit copper at a higher rate at the bottom of the trenches compared to the sidewalls. This disparity in deposition rates ensures that the trenches and vias are filled void-free and seamlessly, even when dealing with high aspect ratios. The Damascene process brought about a significant revolution in interconnect technology by enabling the widespread adoption of copper as the primary metal in advanced integrated circuits, thanks to its superior conductivity and other desirable properties. Finally, integrating low-resistivity metal with low-*k* dielectric reduces resistive–capacitive (*RC*) delay, cross-talk noise, and power consumption in integrated circuits [6].

At present, plasma etching, particularly reactive ion etching (RIE), stands out as the method best suited to meet the demanding requirements of semiconductor fabrication, especially for achieving highly anisotropic patterning. When it comes to SiO_2_ layers deposited using PECVD processes, the choice of plasma precursors is crucial. Typically, an optimized mixture of volatile fluorocarbons is employed. These volatile fluorocarbons play a critical role in generating fluorine atoms during plasma interaction with SiO_2_. The fluorine atoms then react with the silicon dioxide to form volatile etch byproducts. Notably, the natural etching of SiO_2_ by fluorine atoms at room temperature occurs mainly with desorption of SiF_4_ and oxygen, but the reaction rate is quite low and is described by the first-order kinetic equation *R_F_*_(*SiO*2)_ = (6.14 ± 0.49) × 10^−13^*n_F_*∙*T*^1/22^∙exp(−0.163/*kT*) [A/min], with the reaction probability *ε_F_*_(*SiO*2)_ = 0.0112 ± 0.0009∙exp(−0.163/*kT*) [51]. However, when the SiO_2_ is subjected to ion radiation within the plasma, this process is greatly accelerated [52]. The key principle underlying anisotropic etching of SiO_2_ lies in the disparity between the rates of spontaneous etching and ion-induced etching. This difference is what enables the creation of highly directional, anisotropic patterns in the SiO_2_ layer during plasma etching processes. When fluorocarbon precursors are used, fluorocarbon polymers are deposited on the sidewalls of the etched structure, providing additional protection against lateral etching. The concentration of fluorine, carbon, and hydrogen in the precursors is crucial for achieving the optimal ratio of etching and polymer deposition. These concentrations can be adjusted based on the composition of the OSG, the intensity of ion bombardment, and the nature of the bottom etch stop layer.

OSG low-*k* materials have a matrix similar to SiO_2_ but are doped with 10–20 mol% of carbon. The carbon groups impart hydrophobic properties and reduce the density of the SiO_2_ matrix. In addition to this intrinsic lower density, additional artificial porosity is generated by removing sacrificial porogen, which is deposited simultaneously with the matrix. The resulting material becomes porous, which leads to a low dielectric constant (Figure 1). As already mentioned, the etch rate of SiO_2_ by fluorine radicals is relatively low and strongly depends on ion bombardment, which induces lattice damage and increases the quantity of fluorine atoms adsorbed onto the SiO_2_ surface. The quantity of adsorbed fluorine is expected to be a fraction of a monolayer on an annealed surface, a monolayer on a damaged surface, and several monolayers on surfaces where mixing or other synergistic effects are operative. The major reaction products during the interaction of SiO_2_ with fluorine (SiF_4_, oxygen, and oxyfluorides) are formed in this layer.

Methyl groups in OSG low-*k* dielectrics introduce additional challenges due to the low efficiency of etching organic groups by fluorine atoms [53]. The etch rate of OSG is very low in pure fluorocarbon plasma. However, it increases with the addition of oxygen due to the formation of volatile carbon oxides [38]. Although ion bombardment can complicate the etch rate, it offers benefits by densifying the etched surface, which partially seals the pores. This decrease in porosity reduces the likelihood of active species penetrating the low-*k* materials [21].

The porosity of the films also impacts the etch rate and mechanism. Standaert et al. [53] studied the etching of xerogel films with various pore sizes and porosities. It was assumed that the etch rate should change according to the density of the material, following a simple law:*ER_norm_* = (1 − *P*)·*ER*,(4)
where *ER_norm_* is the normalized etch rate, *ER* is the etch rate, and *P* is the porosity. However, the actual etch behavior is more complex. For small pore sizes and 30% porosity, the xerogel etch rate is only partially enhanced by the porosity, as expected according to Equation (4). In plasmas characterized by minimal polymerization, such as CF_4_ or oxygen-rich fluorocarbon plasmas, an additional enhancement is observed, with a factor of up to 1.6. A similar increase is also noted in polymerizing CHF_3_ plasma, where the formation of fluorocarbon film on the surface is relatively low. When the polymerization of the discharge is increased, XPS analysis reveals that fluorocarbon polymers are deposited inside the pores on the xerogel pore surface. At this stage, the xerogel etching is suppressed, and the etch rate, after porosity correction (Equation (4)), falls below the SiO_2_ etch rate. This suppression is more pronounced for xerogel films with higher porosity and larger pore sizes.

Rakhimova et al. [54] investigated the interaction of OSG low-*k* films with atomic fluorine. To understand the mechanism, they excluded the effects of carbon fluorides, ions, and VUV photons by using SF_6_ as a downstream plasma source and a specially designed experimental chamber. It was demonstrated that fluorination of the pore walls is the fastest process of OSG modification, occurring without an activation barrier. Higher porosity and greater pore connectivity promote deep F penetration into the material and fast fluorination. The subsequent slower stage involves H atoms being abstracted by F atoms from −CH_3_ groups, leading to the formation of CF*_x_*H*_y_* surface species. The combined random walk and kinetic model results in the evolution of chemical modification inside the OSG films depending on the F atom dose. The effective etch probability of ultralow-*k* (ULK) SiOCH materials (interacting with a SiO_2_-like matrix) was calculated per one F atom. It was found to be close to the etch probability of dense SiO_2_ at *P* < 15%, then it significantly increased when porosity increased up to *P* ≈ 30%.

Hence, the etching process of porous OSG materials in fluorocarbon plasma entails a complex mechanism. Apart from considering the distinct etch rates of the SiO_2_ matrix and carbon-containing components, one must also take into account the complex process of fluorocarbon polymerization. Nonetheless, the plasma etch recipes for OSG low-*k* films have generally been understood and established. A thorough analysis of various recipes, conditions, and challenges can be found in the provided reference [38]. Presently, a critical issue revolves around plasma damage as it determines the quality and operational aspects of the final integrated devices. This concern was addressed in greater detail in the review paper [38]. One can observe that modern interconnect technology requires different etching approaches for various tasks and applications. To address this need, various modifications of plasma reactors have been developed and utilized (see Figure 4). The right picture in Figure 4 also illustrates the Si, C, and O profiles of low-*k* samples exposed to different etching conditions [55]. Although the results were obtained using an ICP chamber with only bottom power (BPO), top power (TPO), and mixed (T&BP) conditions, they represent conditions typical for CCP downstream and mixed regimes. These images were obtained using energy-filtered transmission electron microscopy (EFTEM). It is evident that the dark areas, indicating the formation of a carbon-free layer (plasma damage), are localized near the top surface in the case of CCP conditions, while the depth of damage extends throughout the entire film in the downstream condition. The mixed (T&BP) condition exhibits a degree of damage similar to CCP, but with a higher etch rate, demonstrating the effect of pore sealing by ion bombardment in CCP conditions. Moreover, it is apparent that the etching process can be efficiently optimized by employing the mixed (T&BP) conditions.

Posseme et al. [56] studied the etch rates of SiOCH, SiO_2_, and SiCH in a medium-density fluorocarbon plasma (CF_4_/Ar/N_2_). It was established that the etching of SiOCH and SiCH materials is controlled by a fluorocarbon interaction layer formed on top of the dielectrics. The etch rate of the dielectrics is significantly influenced by the thickness of this fluorocarbon layer. As the thickness of the fluorocarbon layer increases, less ion energy is dissipated in the bulk dielectric material, leading to a decrease in the etch rate. Additionally, the fluorine content in the fluorocarbon layer impacts the etch rate; a decrease in the fluorine content results in a lower etch rate because there is less free fluorine available to interact with the dielectric interface. During the steady-state etching regime of SiCH and SiOCH films, a fluorinated layer (SiCF and SiOCF, respectively) forms at the interface between the low-*k* material and the top fluorocarbon (CF*_x_*) layer, serving as the fluorine source for the etching reactions. The formation of the fluorocarbon layer is influenced by the plasma operating conditions, such as the source power injected into the plasma source, pressure, gas flow in the etch chamber, and the chemistry used, particularly the polymerizing properties of the gas. The addition of highly polymerizing gases such as C_4_F_6_, C_4_F_8_, or CH_2_F_2_ to CF_4_/N_2_/Ar generates a thicker fluorocarbon layer without altering the F/C ratio of the polymer. Conversely, diluting Ar in the CF_4_/Ar gas mixture results in the formation of a fluorocarbon layer on SiOCH and SiCH with reduced fluorine content. The reason for this effect is likely the increase in electron temperature due to the addition of Ar, which in turn increases the fluorine concentration through the reaction of Ar metastables: Ar• + CF_4_ → Ar + CF_3_ + F. The chemical composition of the low-*k* film also strongly influences the formation of the fluorocarbon layer. The oxygen concentration in the film leads to a thinner fluorocarbon layer and a higher F/C ratio on SiO_2_ compared to SiOCH. Conversely, the concentrations of carbon and hydrogen promote the formation of a thicker fluorocarbon layer and a lower F/C ratio on SiCH compared to SiOCH.

### Plasma Damage

Despite meticulous patterning and integration processes, electrical characterization and reliability testing often reveal unexpected issues. This discrepancy can arise due to various factors such as microscopic material defects, interface issues, process variations, or even subtle environmental factors. Advanced techniques such as failure analysis and accelerated testing can help uncover the root causes of these degraded properties. By identifying and addressing the underlying issues, manufacturers can enhance the quality and robustness of their integrated electronic devices.

One of the most important issues is related to the so-called “plasma damage” of the etched low-*k* materials. Plasma damage of OSG low-*k* dielectrics is a complex phenomenon involving both physical and chemical effects. Chemical modifications include relatively macroscopic changes in chemical composition related to the different reactivity of OSG components (mainly SiO_2_-like matrix and carbon-containing groups) with chemically active plasma components. The most pronounced effects are related to the depletion of carbon concentration in oxygen-containing plasma or their fluorination. Reduction in concentration and fluorination of the carbon-containing groups make OSG more hydrophilic, and the subsequent moisture adsorption increases the dielectric constant and leakage current. Many efforts have been directed towards restoring damaged low-*k* materials using different types of silylation agents and chemical/plasma/UV treatments, but full restoration is generally unachievable [38].

The degradation (increase) of the dielectric constant directly correlates with changes in bonding configuration, the formation of a carbon-depleted layer, film shrinkage, and surface densification. The depth of plasma damage is mainly determined by the diffusion of active radicals (O, H, F, etc.) into the pores and the subsequent removal of organic hydrophobic groups. The penetration depth of active radicals depends on pore size, connectivity, diffusion rates, and their recombination probability. In the case of fluorine-based etch plasma, the depth of damage is influenced not only by these factors but also by the etch rate [57]. The depth of etch damage increases when the etching rate is slower than the speed of damage propagation. Therefore, utilizing fast etching recipes can help reduce the depth of damage. Additionally, intensive ion bombardment is beneficial because surface densification reduces the penetration of active radicals into the pores. Capacitively coupled plasma (CCP), especially dual-frequency CCP with controlled flows of radicals and ions, is preferred for patterning low-*k* materials.

A thorough demonstration of the damage features dependent on the type of plasma reactor and the mechanisms was reported by Kunnen et al. [55]. As previously mentioned, the authors utilized an ICP plasma reactor, and the experiments were conducted under three different conditions: top power only (TPO, pure ICP regime), bottom power only (BPO, CCP-like regime), and a mixed regime with both top and bottom applied powers (T&BP). These conditions revealed entirely distinct phenomena. In the BPO condition, similar to a CCP reactor, the low-*k* surface underwent bombardment by energetic ions. Conversely, the TPO condition generated a high concentration of active radicals, with minimal ion bombardment intensity. In TPO, oxygen radicals deeply infiltrated the pores, reaching the film’s bottom and resulting in complete carbon depletion, as can be seen in EFTEM pictures (Figure 4). The qualitative diffusion–recombination models of plasma damage by oxygen radicals have been proposed and analyzed by Safaverdi [58].

Braginsky et al. [59] conducted an extensive study on low-*k* damage caused by oxygen radicals generated in fast-flow RF CCP plasma using a 90% O_2_/10% Ar mixture at a pressure of 10 Torr. The loss probabilities of oxygen atoms through plasma-induced fluorescence were measured and analyzed. Additionally, X-ray fluorescence (XRF) and FTIR spectroscopy were employed to track the evolution of carbon and oxygen concentrations in the films over time exposed to atomic oxygen flux. Then, the removal of CH_3_ groups was simulated using 1-D Monte Carlo simulation, utilizing a simplified structure of regular vertical pore channels to match the porosity of the materials. It was demonstrated that in low-*k* films with pore sizes approaching 1–1.5 nm, the classic diffusion approach for calculating oxygen atom concentrations in nanoporous materials becomes inadequate. A more appropriate description of experimental results can be achieved by employing ideas from random walk theory. The depth of penetration of oxygen radicals is directly proportional to the pore size and inversely proportional to the sum of recombination and chemical reaction coefficients. Ultra-low-*k* materials, characterized by the lowest dielectric constants, inevitably exhibit higher porosity and larger pore sizes, resulting in increased pore interconnectivity. Consequently, active radicals penetrate deeper into the bulk low-*k* material. To mitigate or prevent damage, strategies such as sealing the uppermost layers of low-*k* films, depositing a thin layer impermeable to oxygen atoms on top of low-*k* films, and enhancing chemical modification of pore surface composition to substantially increase the surface recombination rate of oxygen atoms can be employed. Similar strategies are related to the application of pore stuffing by sacrificial polymers (P4 strategy) [60] and protection by condensed reaction byproducts at cryogenic temperatures [61]. This enhancement can also be achieved by increasing the carbon concentration in low-*k* films [62]. It is necessary to mention that alternative diffusion models have also been proposed. Goldman et al. [63] proposed a diffusion-based Deal–Grove type of model. 

The chemical reactions between the oxygen radical and the Si–CH_3_ groups located on the pore wall surface start with the abstraction of H because H_2_C–H is the weakest bond: ≡Si–CH_3_ + O → ≡Si–CH_2_• + OH(5)

Further reaction of ≡Si–CH_2_• with oxygen atoms leads to the complete loss of the methyl group and the formation of surface active sites (≡Si*) that can adsorb water, forming ≡SiH and ≡SiOH groups: ≡Si–CH_2_• + O → CH_2_O + ≡Si*(6)
≡Si• + H(OH) → ≡SiH (≡SiOH)(7)

Reaction of CH_2_O molecules leads to the formation of CO, CO_2_, and H_2_O.

Unlike O_2_-based plasma, H_2_-based plasmas exhibit varied and sometimes contradictory effects on low-*k* films. While some authors report no effect on low-*k* films, others demonstrate that plasma processes enhance the film properties, while still others indicate severe damage. The paper [64] analyzes most of the studies related to the effects of hydrogen plasma. The contradictory conclusions are related to the wide variety of plasma chambers. In the case of ICP and CCP systems, the etched surface interacts directly with ions and radicals from the plasma. In contrast, DSP (downstream plasma) systems might involve only pure chemical interactions. Modern DSP sources commonly utilize microwave technology and are typically positioned away from the wafer area, separated by a specialized grid. This grid functions to counteract electrically charged particles (electrons and ions) and also attenuates the flow of UV/VUV light. Therefore, only the hydrogen atoms interact with the wafer. The application of H radicals, typically generated from downstream H_2_ plasma, is indeed crucial for damage-free processing, especially in the context of cleaning low-*k* surfaces prior to barriers and metal deposition. However, hydrogen plasma can significantly damage OSG low-*k* when it is simultaneously affected by ions and UV light. The lack of damage in the DSP hydrogen process can be elucidated by referencing the findings mainly reported by Han [65], Worsley [64], Lazzeri [66], and Rakhimova [66]. They investigated the interaction between H radicals and the low-*k* surface, highlighting their ability to effectively remove contaminants and impurities without causing detrimental damage to the underlying material. Lazzeri et al. [66] have contributed insights into the fundamental mechanisms governing this interaction, shedding light on how H radicals selectively react with surface species while minimizing adverse effects such as etching or sputtering. By leveraging the knowledge gleaned from these studies, semiconductor manufacturers can optimize their processes to ensure efficient cleaning of low-*k* surfaces while preserving their integrity, ultimately leading to improved device performance and reliability. According to the findings reported by Rakhimova et al. [59], the reaction initiates with the detachment of a hydrogen atom from the ≡Si–CH_3_ group, forming a radical species (≡Si–CH_2_ + H). Subsequently, the reaction of the resulting ≡Si–CH_2_ radical with oxygen atoms leads to its complete destruction, while the reaction with hydrogen atoms restores the original ≡Si–CH_3_ group (≡Si–CH_2_ + H → ≡Si–CH_3_). This mechanism finds support in the results presented by Lazzeri et al. [66]. In their study, they exposed OSG low-*k* films to a deuterium plasma. Deuterium, being an isotope of hydrogen, replaces hydrogen in the low-*k* matrix. However, the total concentration of hydrogen and deuterium (H + D) remains constant throughout the process. This phenomenon suggests that the dangling bonds formed by hydrogen detachment from the low-*k* films are saturated by the reaction with deuterium, leading to the formation of ≡Si–CH_2_D species (≡Si–CH_2_ + D → ≡Si–CH_2_D), while the sum of (H + D) remains constant. UV light and ions promote the direct detachment of CH_3_ groups from Si and make this reaction irreversible. 

Nitrogen and ammonia are often used during the plasma etching/stripping and metal barrier deposition (TaN, Ta, MnN, AlN, …). The addition of nitrogen to a hydrogen plasma significantly influences low-*k* damage, despite the fact that N radicals alone have minimal impact on low-*k* materials. However, the presence of a combination of N_2_ and H_2_ radicals, along with ammonia plasma, has a detrimental effect on low-*k* films. One possible hydrophilization mechanism involves replacing Si–CH_3_ bonds with hydrophilic Si–NH_2_ bonds, which can subsequently be transformed into Si–OH bonds through hydrolysis with ambient moisture. The formation and existence of Si–NH_2_ bonds have been confirmed by FTIR. The reduction in carbon content in low-*k* materials may also be attributed to the formation of volatile HCN molecules [67,68].

VUV light generated in plasma plays an important role. VUV photons can damage OSG low-*k* materials by partially or completely destroying Si–CH_3_ bonds, resulting in hydrophilization and the formation of electrically active dangling bonds [69,70]. However, the most presently important modifications are related to the generation of electrically active defects that cannot be identified by simple chemical analysis, such as FTIR spectroscopy.

The most important impacts of UV light on OSG dielectrics are related to change in dielectric constant, built-in charges, leakage currents, and breakdown field. Although the SiO_2_-like skeleton represents the essential part of the low-*k* materials, the presence of significant amounts of alkyl groups and organic residues, as well as porosity, brings numerous novel aspects to the UV/VUV response of these layers. VUV exposure can cause demethylation of OSG films (especially in the presence of active gases: O_2_, NH_3_) [71,72], making them hydrophilic and leading to changes in electrical characteristics.

The exposure of amorphous SiO_2_ insulators to VUV light results in the accumulation of a fixed charge. This process is associated with trapping of photogenerated (or injected from electrodes) mobile charge on the pre-existing oxide defects. If the photon energies exceed the bandgap width of the oxide (8.9 eV for SiO_2_) [73,74], positive charging caused by hole trapping is dominant [75,76,77,78]. Negative charge buildup can also be observed upon electron photoinjection and trapping in SiO_2_, but with a much lower trapping rate than the positive charging upon hole injection [79]. Generation of energetically deep states in the oxide bandgap, facilitating leakage current and leading to dielectric breakdown, is the most relevant for OSG low-*k* dielectrics.

The depth and degree of plasma damage strongly depend on the wavelength of VUV light. It has been shown that the light with a wavelength shorter than 200 nm has sufficient energy to break Si–CH_3_ bonds [25]. Efficiency of the bond breaking increases with decreasing wavelength, but the overall degree of damage at very short wavelengths (<150 nm) can be smaller due to the high absorption coefficient and limited depth of light penetration. This is the reason why the most damaging wavelengths correspond to the range close to 150 nm when photons have sufficient energy to break Si–CH_3_ bonds, and the depth of light penetration is still higher than the typical film thickness (close to ±100 nm in modern interconnect technology) (Figure 5) [80,81,82,83].

Figure 5 shows that the experimentally measured integral carbon depletion is maximal when the film was exposed to VUV light with a wavelength of 147 nm. The VUV photons in this case are already sufficient to break the Si–CH_3_ bond, and the absorption coefficient is still quite low, allowing the light to penetrate through the 100 nm thick films. Further wavelength reduction generates the photons that are also able to break Si–CH_3_ bonds, but the depth of damage is smaller because of the high absorption coefficient. So, VUV photons generated by Ar and He plasma (106 and 58 nm respectively) cause less damage than 147 nm (Xe plasma) and 13.5 nm (Sn^7+^ plasma used in EUV lithography). The most damaging plasma is Xe (147 nm), but light with a similar wavelength can also be generated by CF_2_ radicals [84], which are common components of various etch recipes utilizing different fluorocarbon precursors.

Another important characteristic of VUV damage is its quantum yield of CH_3_ group abstraction that depends on the material’s properties. Dependence of the quantum yield on porosity was measured by Lopaev et al. [83] using a few different methyl-terminated OSG films (Figure 6). It is clear that there is a very strong dependence on porosity, and the curve has percolation-like character. The quantum yield drastically increases at porosity close to 45%.

Unfortunately, such data were generated mainly for methyl-terminated OSG materials, and the information is very limited in the case of PMO-like OSG materials with a carbon bridge in their matrix. Some general ideas about the resistance of these materials can be demonstrated by the results of quantum chemical calculations (Figure 7). This figure shows the methylene-bridged PMO OSG film with the simultaneous presence of methyl terminal groups. The VUV-induced excitation of this molecule at 12 eV initially occurs in the singlet state S^n^. The dissociation energy (*E_diss_*) of a potential bond breaking in the model molecule was calculated as the difference between the free Gibbs energies of the molecule in the ground state and the dissociation products. After excitation into the electronically excited S^n^ state and relaxation into the first excited singlet state S^1^, followed by intersystem crossing (ISC), the triplet state was formed. This triplet state has sufficient energy to undergo different bond scissions.

One can see that the chemical bonds present in OSG films containing both bridging and terminal carbon can be separated into two different groups from the point of view of dissociation energy. As expected, the weakest chemical bonds (*E* < 85 kcal/mol) are represented by Si–C bonds from Si–CH_2_–Si bridge (R1 and R4) and the bonds between the silicon atom and the terminal methyl group (R2 and R3). The detachment of hydrogen atoms also occurs relatively easily (R5 and R6). One can see that the difference between R1 and R4, R2 and R3, and R5 and R6 reflects the influence of neighboring groups and is easily understandable. For example, R1 < R4 and R8 < R9, since the central Si atom is bonded to the hydroxyl (R10) and, therefore, has a positive charge that enhances the bonding energy with the neighboring CH_2_ group and the oxygen atom (R9). The bond R1 < R2 and R3, indicating that the Si bond with the bridging methylene group is the weakest in this molecule. The second group of chemical bonds has dissociation energy >100 kcal/mol and includes Si–O bonds (R8 and R9), detachment of the hydroxyl group (R10), and detachment of an H atom from the hydroxyl. 

The low VUV resistance of bridging carbon groups was experimentally confirmed in the paper [42]. Moreover, it was shown that the benzene bridge has lower stability for light with λ ≥ 200 nm compared to methylene and ethylene bridges. The optical light absorption characteristics play an important role, and this will be demonstrated in the discussion below.

One crucial aspect to note is that when Si–C (SiCH_3_) bonds break, they leave a dangling bond on the Si atom. The subsequent behavior of this bond can vary. It may become saturated with hydrogen atoms, forming Si–H bonds through reactions with hydrogen atoms produced from detached CH_3_ radicals or water molecules. Additionally, the formation of oxygen-deficient centers (ODC) like ODC(I) and ODC(II) (Figure 8) is conceivable. However, in the case of low-*k* materials, the temperatures used are constrained by BEOL requirements, preventing the matrix from relaxing and greatly limiting the likelihood of such reactions. Furthermore, Marsik et al. demonstrated that there is an anticorrelation between the removal of CH_3_ and the formation of SiH groups during UV curing. This suggests that nearly all Si dangling bonds are saturated by hydrogen atoms generated from the destruction of desorbed CH_3_ groups.

As already mentioned, VUV light degrades the electrical characteristics of low-*k* materials [69,85,86,87]. The VUV photons from the processing plasma increase the intrinsic defect density and create trapped charge inside the low-*k* material [87,88]. During the ion sputtering process, atoms can be knocked off from the low-*k* material network, leading to the formation of Si vacancies, such as EX centers [89,90] or dangling carbon bonds [69,87,91,92,93,94,95,96]. These carbon-related defects contribute to increased leakage [97]. In addition, the formation of surface oxygen-deficient centers (vacancies) on the pore wall may lead to the formation of sub-gap surface states at 5.0 and 7.2 eV. Atomic defects such as non-bridging oxygen hole centers (NBOHCs) and oxygen vacancies (E’-centers) have extensively been studied by electron spin resonance (ESR) spectroscopy (Figure 8) [25,96].

ESR spectroscopy allows us to identify only paramagnetic defects like E’, POR, and NBOHC. Meanwhile, in the case of SiO_2_, important contributions to electrical characteristics are made by diamagnetic oxygen-deficient centers ODC(I) and ODC(II). Normally, they can be identified using UV-induced luminescence and characterized by emission peaks with energies of 3.1 and 4.3 eV (ODC(I)) and 2.7 and 4.4 eV (ODC(II)). The luminescence from differently deposited SiO_2_ layers has been reported in many papers, and these peaks are always attributed to ODC [25,96]. However, most of the results are related to SiO_2_ layers fabricated at high temperatures.

Recently, the UV-induced photoluminescence (PL) of a mesoporous organosilica low-*k* dielectric with an ethylene bridge was studied [98], and the observed peaks were interpreted as related to the formation of oxygen-deficient centers ODC(I) (≡Si–Si≡) and ODC(II) (=Si:) centers (Figure 8f,g), similar to those observed in pure SiO_2_ [99,100]. It was assumed that these centers can be correlated with the character of leakage current studied in ref. [101]. However, another recent luminescent study, based on the evaluation of various OSG dielectrics with different porosity and chemical composition [102], demonstrated that the origin of the observed luminescent bands can be related to the film’s components rather than solely to the presence of oxygen vacancies. Therefore, it is not always straightforward to explain the degradation of electrical properties with the formation of oxygen vacancies.

There are several factors that differentiate the response of low-*k* insulators to UV/VUV exposure from that of *a*-SiO_2_ [25]. First, most of the SiO_2_ films reported in the literature with reliable identification of these defects are high-temperature versions of amorphous SiO_2_, synthesized either by thermal oxidation of silicon or from a synthetic silica melt. These materials possess a relaxed network structure with a relatively narrow distribution of Si–O–Si angles around an average value of 144°. In contrast, the processing temperature for low-*k* materials is limited by BEOL (back-end-of-line) requirements to below 450 °C, which is too low to allow the network to relax. This factor results in a high concentration of network configurations with extreme bonding angles, which are expected to be more prone to chemical reactions [103], including those with hydrogen released under UV/VUV illumination conditions from electrodes of the low-*k* material itself. The limited thermal budget is especially a key issue for sol–gel-based films. Sol–gel chemistry uses stable precursors, and their polymerization is only related to hydrolysis and further condensation with the formation of Si–O–Si or Si–R–Si bridges and methyl groups located on the pore wall surface. Potentially, if Si–CH_3_ bonds are broken by VUV light, the formed E’-defects can be considered as potential precursors for the formation of ODC centers. However, at temperatures below 450 °C (the temperature used for curing low-*k* materials), these defects are immobile, and there is insufficient energy for structural relaxation.

Second, the low-temperature SiO_2_-like matrices in OSG low-*k* are prepared in the presence of organic templates, porogen precursors, or by using spin-coating. They are usually OH-rich as opposed to the *a*-SiO_2_ films thermally grown on silicon or fabricated by O-ion implantation into Si crystal. The latter are usually O-deficient and exhibit characteristic ESR signature of this deficiency—the well-known E’-centers [104,105,106,107,108]. This difference can clearly be seen from the ESR spectra (Figure 9) taken from the “conventional” PECVD *a*-SiO_2_ (Figure 9a) and two OSG low-*k* dielectrics: spin-on deposited nano-crystalline silica (NCS) (Figure 9b) and porous UV-cured CVD-processed “black diamond” (BD, labeled as CVD1 throughout this paper) insulator prior (Figure 9c) and after He ion bombardment (Figure 9d) [109]. While the CVD-SiO_2_ shows not only the E’_γ_-line at *g* = 2.0005 with a characteristic powder pattern stemming from dangling bonds of silicon atoms in an *a*-SiO_2_ matrix, but also the 72.5 G doublet associated with the presence of one hydrogen atom in the back-bond of the kernel Si atom, neither NCS nor CVD1 materials exhibit these O-deficiency features. Even after extended VUV (*hν* = 10 eV) exposure, no detectable E’_γ_ signal can be traced in these samples. Similar observations were also made on other low-*k* insulators ranging from spin-on glass to self-assembled dielectric layers [91,92]. Only after sputtering, a “new” ESR signal at *g* = 2.00247, which can be identified as an EX-center representing a Si vacancy in an *a*-SiO_2_ matrix [89,90], clearly points towards O-enrichment in the low-*k* oxide case. Later, high-resolution ESR analysis [88] revealed an additional component of the ion-bombardment-induced signal tentatively associated with the formation of dangling bond defects in oxycarbide clusters. This association was made because a similar signal has also been found in *a*-SiOC matrices.

It has been reported [110,111] that dielectric failure times and charges to breakdown decrease for VUV-exposed low-*k* dielectrics. ESR results from Ren et al. [112] indicate that VUV photon irradiation generates additional defects and twisted bonds in the structure. Other literature reports also discussed increasing defect densities and leakage currents after VUV exposure [69]. Sinha et al. [87] suggested that photon irradiation generates trapped charges that may lead to reliability issues. Afanas’ev et al. [88] studied the nature of the defects generated during ion bombardments using three discharging gases: H_2_, He, and Ar. It was found that ion sputtering causes knock-offs of atoms from the low-*k* material network, leading to the formation of Si vacancies or dangling carbon bonds. These defects contribute to the increased leakage. Furthermore, the results of King et al. [97] show the generation of surface oxygen vacancies, likely resulting from the removal of terminal organic groups after Ar^+^ sputtering, with two related sub-gap surface states observed at 5.0 eV and 7.2 eV. Nichols et al. [111] also studied the effect of ion energy in the plasma reactor and found that the increase in ion energy resulted in higher leakage currents and reduced breakdown fields. They attributed this phenomenon to the formation of an oxide-like layer caused by the loss of carbon near the film surface. It was further suggested that defects are more easily generated within this oxide-like layer. Sinha et al. [87] proposed that ion bombardments cause ions to adhere to the dielectric surface, leading to charge accumulation that negatively impacts reliability.

## 4. Electrical Properties

### 4.1. The Optical Properties and Bandgap of Organosilica Films

An important characteristic of dielectric materials is their bandgap. While the bandgap of porous OSG low-*k* dielectrics featuring terminal methyl groups has been investigated by multiple researchers using diverse methodologies, certain aspects remain inadequately elucidated. Specifically, these unresolved matters pertain to the intrinsic defects’ nature, their source, and their influence on the bandgap as well as electrical properties. The electron energy loss spectroscopy measurements [113,114] determined the OSG low-*k* dielectric bandgap to be 8.5 eV and 10 eV, respectively. The results obtained using reflection electron energy loss spectroscopy [97], ellipsometry [23], VUV spectroscopy [115], and X-ray photoelectron spectroscopy [116] show that the bandgap of porous OSG dielectrics with *k* = 2.0–3.3 ranges between 7.5 and 10 eV. These values are quite close to those of amorphous SiO_2_, which range between 8.0 and 9.0 eV [117,118]. Moreover, the barrier height at the interfaces of low-*k*/metal (Ta, tantalum) and low-*k*/Si was determined to be 4.5 eV according to internal photoemission experiments [109,119]. This also proves that the bandgap of most PECVD OSG dielectrics is similar to that of SiO_2_. It confirms that the carbon in methyl-terminated low-*k* films is not incorporated into the matrix. If the carbon-containing components are present in the network as Si–C–Si-like bridging groups, the bandgap value would drop dramatically. The barrier height at the copper/low-*k* dielectric interface also depends on the amount of network carbon in the film and ranges from 1 to 4 eV [120].

These results allow us to expect that the bandgap of OSG films with carbon bridges incorporated into their matrix can be different. As previously mentioned, the majority of films currently employed in the microelectronics industry are methyl-terminated OSG deposited via PECVD techniques. These materials resemble silica, with certain oxygen atoms in the silica matrix substituted by two methyl groups (≡Si–O–Si≡ → ≡Si–CH_3_ … CH_3_–Si≡) [7]. It diminishes the film’s density and exacerbates the mechanical properties, which are vital for its integration into ULSI devices. The necessity to enhance the mechanical properties and reliability of low-*k* dielectrics has driven extensive development and research into materials featuring various types of carbon bridges between silicon atoms [5,29,121,122,123,124,125]. Substituting the oxygen bridge with carbon leads to enhanced mechanical properties owing to the greater bending rigidity of the ≡Si–C–Si≡ bonds compared to the ≡Si–O–Si≡ bonds. The application of EISA [126] employing carbon-bridged alkoxysilane precursors has allowed the production of PMO with ordered porosity and the formation of hydrocarbon bridges within the film matrix. Extensive evaluations have been conducted on their properties, including the mechanical properties, thermal, chemical, and VUV resistance of various carbon bridges [39,42].

#### 4.1.1. Optical Properties of Various OSG Materials

To the best of our knowledge, the optical properties and bandgap measurements of PMO films with various carbon bridges have not yet been reported in the scientific literature. Therefore, to analyze the expected trends when introducing different carbon groups into the OSG matrix, we conducted quantum-chemical calculations of the optical properties of various OSG materials (Figure 10). The presented data were calculated using the density functional theory (DFT) PBE0-D3/6-31G** level of theory as implemented in the Jaguar 9.6 program [127,128,129,130].

Figure 10a displays the absorption spectra of a SiO_2_ (1a) fragment alongside fragments of methyl-terminated OSG matrices with one (2a) and two (3a) methyl groups (refer to the top line of Figure 11). It is evident that the incorporation of methyl terminal groups barely alters the absorption spectra, maintaining an optical bandgap close to 8.2–8.3 eV. These findings align with the aforementioned experimental results. Conversely, a more pronounced change in the optical bandgap is observed with the introduction of bridging carbon-containing groups (Figure 10b). The films with methylene and ethylene bridging groups exhibit values close to 6.8–6.2 eV for the optical bandgap. Structures featuring benzene bridges demonstrate a more significant reduction in the bandgap, reaching values close to 5.5 eV (hyperconnected structure) [125] and 5.0 eV (linear bridge).

#### 4.1.2. Change of Optical Characteristics during UV Curing

Figure 12 shows the evolution of the optical characteristics of PECVD OSG films with methyl terminal groups after broadband UV-assisted thermal curing (λ ≥ 200 nm, *T_a_* = 430 °C in nitrogen) measured using UV ellipsometry. One can see that the index of refraction and the absorption coefficients are changing simultaneously. Change of refractive index is related to change of porosity because of porogen removal. The full porosity was calculated using the Lorentz–Lorenz equation:(8)$$P=1-\left(\frac{n_{0}^{2}-1}{n_{0}^{2}+2}\right)/\left(\frac{n_{m}^{2}-1}{n_{m}^{2}+2}\right),$$
where *P* is porosity, *n*_o_ and *n_m_* are the indices of refraction of porous films and matrix, respectively. The results of porosity calculations for films with different curing times are shown in Figure 13. The graphical representation unequivocally demonstrates that films attain maximal porosity at the optimal curing time, denoted as T, which depends on the temperature and UV light (both intensity and wavelength) and is usually about a few minutes. At 0.2 T and 0.5 T intervals, porosity remains comparatively lower due to the remaining porogen within the pores. Conversely, overcuring instances (2 T and 5 T) lead to diminished porosity as a consequence of micropore collapse. Notably, the absorption coefficient’s behavior warrants scrutiny. Instances of suboptimal curing durations (0.2 T and 0.5 T) exhibit conspicuous peaks around 6.5 eV, with the additional presence of a peak at 4.5 eV in films subjected to 0.2 T. Overcuring phenomena (2 T and 5 T) notably diminish the prominence of the 6.5 eV peak, while accentuating the peak at 4.5 eV. The nature of these peaks in OSG films was investigated and clarified by Marsik et al. [23,131]. They aimed to comprehend the origin of absorption within the 4 to 8 eV range. The dedicated experiments involving the deposition of pure porogen (*α*-terpinen) onto a clean Si wafer without the low-*k* material have been carried out. Subsequently, they scrutinized the spectral evolution under UV exposure conditions mirroring those encountered during the curing of OSG low-*k* films. Indeed, the experiments conducted with pure porogen facilitated the inference that the observed absorption peaks within the 4–8 eV range in OSG films may not be attributable to defects within SiO_2_, as is commonly interpreted [99,132]. Marsik et al. observed identical peaks and demonstrated that the sp3 carbon originating from pristine porogen transforms into sp2 carbon under VUV radiation, and they are also discernible in FTIR spectra, indicating the formation of C=C bonds. These bonds are construed as porogen residue—the carbon-rich residue formed due to the dehydrogenation of the porogen polymers. It is noteworthy that the most intensive formation of this residue occurred when light with wavelengths shorter than λ ≤ 190 nm (e.g., λ = 172 nm) is utilized. This residue detrimentally affects the dielectric properties of OGS films, elucidating why broadband UV light with wavelengths exceeding 200 nm is commonly preferred, notwithstanding its comparatively lower kinetic efficiency. The same peaks also manifest during the curing of porogen-based OSG low-*k* films. Consequently, it can be inferred that these peaks are associated with the porogen, with the peak at 6.5 eV corresponding to sp3 carbon (pristine porogen), and the peak at 4.5 eV corresponding to sp2 carbon (referred to as porogen residue—an amorphous carbon-like residue that significantly affects leakage current). Optimizing the curing process is crucial, as repeatedly suggested, considering the evident adverse effects of overcuring [133]. However, entirely preventing the formation of porogen residue in a standard low-*k* film fabrication procedure (deposition → UV-assisted thermal curing) is likely not feasible. An alternative technology proposed by Urbanowicz allows for the fabrication of films without any porogen residue [134]. This technology is based on the porogen removal by atomic hydrogen before UV-assisted thermal curing, and the properties of the obtained films will be discussed later.

The data presented in Figure 12c corroborate the conclusion that the emergence of absorption peaks within the 4–8 eV range in the investigated OSG films is associated with the presence of porogen and its dehydrogenated residue. The film with *k* = 3.0 was deposited without a porogen and exhibits only open porosity of about 6% due to the presence of methyl terminal groups. It can be observed that there are no peaks at 6 and 4.5 eV in the absorption spectra. However, these peaks appear in the films with *k* = 2.5 (*P* = 23.5%) and *k* = 2.3 (*P* = 38%) because these films were deposited with a porogen. Furthermore, the film with *k* = 2.3 was deposited with a higher porogen concentration and therefore has a more pronounced concentration of porogen residue. It is important to observe that upon comparing the experimentally obtained absorption spectra depicted in Figure 12 with the theoretical spectra shown in Figure 10, the experimentally evaluated OSG films exhibit an extended absorption tail reaching up to 2 eV that was not visible in the calculated spectra. This indicates that, besides the clearly defined sp2 and sp3 carbon structures, the materials also contain additional carbon residues with indistinct compositions. Subsequent discussions will demonstrate that analogous conclusions were drawn based on other experimental analyses.

#### 4.1.3. Defect States in the Bandgap of Methyl-Terminated OSG Films

The optical properties of intrinsic defects in SiO_2_ are well studied and documented in different publications and summarized, for instance, in the review papers by Skuja [99] and Griscom [132]. Building upon the concepts outlined in previous studies and anticipating the similarity between silicon dioxide (SiO_2_) and OSG, King et al. [97] conducted a comprehensive examination of the potential formation of ODCs. This investigation utilized reflection electron energy loss spectroscopy (REELS) to provide detailed insights. They measured the bandgap and energy of sub-gap defect states for both non-porous and porous low-*k* materials deposited by PECVD. The dense low-*k* (*k* = 2.8–3.3) was deposited at 400 °C, while the porous films (*k* = 2.3) were deposited at 280 °C. Then, all these films were UV-cured at 400 °C. The measured bandgap of the non-porous low-*k* was approximately 8.2 eV. Ar^+^ sputtering of the non-porous film created sub-gap defect states at ~5.0 and ~7.2 eV. The porous low-*k* shows a slightly smaller bandgap (7.8 eV) and a broad distribution of defect states ranging from 2 to 6 eV. These defect states were attributed to a combination of both oxygen-deficient centers (5.0 and 7.2 eV) and carbon-based porogen residues (2–6 eV). One can see that all these observations are similar to the data presented in Figure 10 and Figure 12, with the only difference being that the separate peaks in our work were 4.5 and 6.2 eV, and they were assigned to sp2 and sp3 coming from porogen residue (Table 3). The obtained information can be plotted as band alignment for OSG low-*k*/barrier structure (Figure 14). One can see that the defect states related to the presence of ODCs and sp2/sp3 carbon are located very close to each other and have significant overlap, making it very difficult to distinguish their impact on electrical characteristics.

#### 4.1.4. Effect of Porosity on the Bandgap

Dependence of the bandgap on porosity and its degradation after ion sputtering was reported in ref. [135]. Using PECVD-deposited methyl-terminated OSG films with different porosity, the bandgap was measured by using core-level X-ray photoelectron spectroscopy (Table 4). The bandgap of pristine samples is almost independent of porosity. Sputtering of low-*k* surface by ion bombardment (Ar^+^ ions 4 keV, 15 mA/cm^2^, 30 s) reduces the bandgap from 8.0–8.3 eV to 6.1–6.8 eV depending on porosity. Changes in chemical composition and defects generation were studied using XPS and ESR spectroscopies. It was concluded that ion-induced defects in OSG low-*k* materials are oxygen/silicon vacancies, similar to the oxygen-deficient centers (ODCs) and silicon dangling bond centers found in bulk SiO_2_, along with carbon dangling bonds originating from carbon silicide clusters within the material. The predominant peak in ESR spectra is characterized by a *g*-factor of 2.0006 and a linewidth of 2 G. These have been attributed to surface oxygen vacancies (SOV), in which the defects are usually positively charged in the paramagnetic state [94]. The other signal at *g* ≈ 2.0026 with an approximate linewidth of 7 G is identified as related to carbon dangling bonds (CDB) back-bonded to C or Si atoms [88].

A similar tendency of bandgap reduction with porosity was demonstrated by Van Besien [136] (Figure 15), who studied several PECVD low-*k* films deposited using the same matrix precursors and the porogen, and also in the papers published by Grill [137,138]. All samples were thermally cured at 430 °C with broadband UV light (λ > 200 nm). The electrical characteristics were measured by using a test structure based on metal–insulator–semiconductor (MIS) planar capacitors [139].

As previously discussed, surface oxygen vacancies have been identified as potential factors contributing to the formation of sub-gap surface defects, observed at approximately 5.0 and 7.2 eV. These defects reside in the upper portion of the bandgap and are situated near the conduction band minimum (Figure 16), while the silicon dangling bonds possess a highly localized state near the mid-bandgap, lying near the isolated sp3 or π hybrid energies, which will not contribute to the states below the Fermi level [140]. As a result, additional electron states near the valence-band maximum most probably can be attributed to the carbon-related defects created during ion sputtering, which can give rise to “deep” energy levels near the valence band in the bandgap of the SiO_2_-like skeleton. For example, the excessive carbon dangling bonds at the interfaces result in electron states distributed in the lower half of the oxide bandgap; also, electron states associated with CH*_x_* layers can be found in the same energy range [141,142].

On the other hand, the bandgap narrowing is actually related to the amount of carbon in the films. Grill has reported significantly reduced optical bandgaps (*E_opt_*) ranging from 3 to 6 eV for low-*k* SiOC:H dielectrics [137,138]. These measurements, based on spectroscopic reflectance, clearly showed a correlation between *E_opt_* and the carbon content present in the SiOC:H films.

The largest bandgap reduction of 2.2 eV occurred in those *a*-SiCOH films (*k* = 2.2) that were deposited with the highest porogen concentration and exhibited the highest concentration of carbon dangling bonds. This is supported by the highest intensity ESR signal (*g* = 2.0026) after ion sputtering. Hence, it seems logical to conclude that the carbon-related defects have the most important contribution to the bandgap narrowing in porogen-based OSG films (Figure 16).

### 4.2. The Leakage Current

The change in leakage current in porous OSG low-*k* dielectrics is more complex compared to dense dielectrics. The primary reason for this complexity is their porous structure. The electrical conductivity significantly depends on porosity because the pore wall surfaces can accumulate various conductive impurities, which influence the observed phenomena. Only careful fabrication of these films, preventing the accumulation of impurities on the pore walls, allows for an accurate analysis of the leakage current mechanisms through the low-*k* matrix.

#### 4.2.1. Effects of Porosity, Porogen Residue, and Adsorbed Moisture

Figure 17 depicts the change in leakage current versus the applied electric field in different low-*k* films (Table 5). Accurately gathering data to reveal the precise mechanism of leakage current poses a complex challenge. A special structure known as the planar capacitor (p-cap) was prepared and used for these measurements [139,143]. The p-cap offers distinct advantages over other simpler quick-turn monitors (QTMs) due to its hermetically sealed environment, which safeguards the test material from damage during electrical probing. Consequently, p-caps serve as precise QTMs for assessing new materials, streamlining the process with fewer steps, and facilitating quicker access to information.

It is evident (Figure 17) that the leakage current of the most porous CVD3 films increases at the lowest applied electric field (Table 5). This increase in leakage current can be attributed to the highest porosity generated in the film, achieved by depositing it with the highest porogen concentration. This concentration left behind the highest amount of porogen residue after UV-assisted curing [144].

Porogen residue (sp2-hybridized carbon) is formed during UV curing, and it is deposited on the pore wall surface. It increases the leakage current and decreases the breakdown voltage of low-*k* materials. The amount of porogen residue increases with the increasing porosity of PECVD low-*k* films due to the higher internal surface area and a larger amount of co-deposited porogen. The electrical characteristics of PECVD ultra-low-*k* (ULK) films are significantly worse compared to organic low-*k* materials and SOG low-*k* films prepared without porogen by self-assembling of nanocrystalline silica because they were deposited without porogen (Figure 17b). Wu et al. [145] showed that the leakage current in this low-*k* dielectric remains constant regardless of temperature within the high electric field range. A barrier height of 4 eV at the low-*k*/metal (Ta) interface is calculated using the theory of FN tunneling.

Another surface compound that increases the leakage current is adsorbed water. Plasma damage or non-optimized UV curing can lead to the partial reduction of methyl terminal groups. The resulting dangling Si bonds can then adsorb water molecules or become saturated by hydrogen atoms formed due to the destruction of methyl radicals. The formed SiH groups are also hydrophilic and can accumulate adsorbed water, which is clearly visible in FTIR spectra and has been demonstrated numerous times. Removing adsorbed water is challenging because different adsorbed forms require annealing at temperatures exceeding 600 °C, which is comparable to the thermal stability of low-*k* materials themselves [21]. As a result, complete thermal restoration of the electrical characteristics of degraded low-*k* films is almost impossible.

The results reported above suggest that the pore wall surface and its contamination play a crucial role in the leakage current. This statement was reported in many publications [6,7,10,21] and is clearly demonstrated in Figure 18 [146]. Using different curing procedures and consecutive chemical analyses, Krishtab et al. [146] demonstrated that the most significant factor influencing the leakage current is adsorbed water, followed by the template (porogen) residue. However, their relative impact obviously depends on the concentration of these impurities. Efficient removal of adsorbed water and porogen residue leads to achieving minimal leakage current.

This conclusion is in agreement with the results obtained by Vanstreels et al. [147], who reported quite different observations. They studied porogen residue-free films using the technology reported by Urbanowicz [148]. After co-deposition of matrix material and porogen, an H_2_ downstream treatment was applied with a special setup to avoid VUV exposure of the low-*k* surface before UV curing. Hydrogen atoms transform the porogen chains into volatile molecules, enabling the complete removal of the porogen without breaking the Si–CH_3_ bonds. This results in residue-free low-*k* dielectrics with enhanced mechanical properties attributed to a more cross-linked network. The uniqueness of these films lies in the complete removal of porogen by hydrogen atoms before UV curing, rendering them completely porogen residue-free. Moreover, the films are highly hydrophobic, allowing the exclusion of the impact of adsorbed moisture. It has been reported that in these films, with porosity ranging from 30% to 50%, the leakage current is independent of porosity (Figure 19), leading to the conclusion that the pore wall surface no longer plays a key role. In this case, the leakage current is defined by different factors, possibly including the presence of ODC centers, and is much lower than in ordinary porogen residue-containing PECVD films. This fact suggests that the influence of all conducting factors other than porogen residue and adsorbed water is much lower. Chen Wu also studied these films with extended porosity range of 10–45% [149]. The tunneling current of films with higher porosity, as described by the FN model, was significantly lower than in PECVD films prepared using traditional technology. This indicates a better matrix quality with a lower defect density. FTIR analysis results validate that films with high porosity exhibit a greater number of network bonds.

Therefore, the conductivity becomes independent of porosity and primarily depends on defects in the matrix. Here, ODC can be considered as possible candidates defining the leakage current. However, the mechanism of their formation is completely different than in high-temperature oxides due to the very low thermal budget dictated by interconnect technology requirements. The temperatures involved are much lower than those typically needed for matrix relaxation (≥1000 °C), which is a key requirement for ODC formation. A possible mechanism of their formation can be related to the removal of methyl groups from micropores that collapse during the overcuring (Figure 13) [133]. In this case, the silicon atoms’ dangling bonds formed by CH_3_ groups’ removal are located at a short distance due to the negative curvature of micropores. The increase in reactivity of chemical groups (hydroxyl groups) in pores with a small diameter has already been studied experimentally and discussed [150,151].

Other theoretical models have also been developed to understand the effect of porosity. Kayaba et al. [152] calculated the magnitude of electrical fields in porous SiCOH dielectrics and demonstrated that the highest field strength corresponds to the air/skeleton interfaces. Hong et al. [153] utilized finite element and Monte Carlo simulations to investigate the electrical field and conduction mechanisms within porous low-*k* films with varying shapes and interconnectivities of pores. The local enhanced fields increase with porosity, leading to poorer insulating properties. Lee et al. [154] also found similar results, suggesting that pore-related field enhancement facilitates the movement of charged species and bond breakage.

Ogawa et al. [155] utilized the percolation model to understand the influence of porosity in low-*k* dielectrics. In this model, dielectric breakdown occurs when a single column of defective cells forms between electrodes. It is shown that pores reduce the time required to generate a percolation path, decrease the breakdown strength, and lower the Weibull shape parameter [156].

#### 4.2.2. Leakage through the Low-*k* Matrix

In OSG films prepared without porogen residue and adsorbed moisture, the conductivity depends on the matrix properties [147]. According to current concepts [157], the conduction mechanisms of dielectrics are classified into contact-limited and bulk-limited types. Contact-limited mechanisms include the Schottky effect [10], thermally facilitated tunneling at the contact [158], and the Fowler–Nordheim effect [159], as illustrated in Figure 20a–c [160,161].

The Schottky effect involves the lowering of the potential barrier at the metal–dielectric boundary due to image forces, as shown in Figure 20a. This effect is observed at high temperatures, in weak electric fields, and when the barrier values are small. Thermally facilitated tunneling at the contact, illustrated in Figure 20b, occurs when an electron transitions to an excited state due to thermal energy and subsequently tunnels into the conduction band. The Fowler–Nordheim effect, depicted in Figure 20c, occurs when an electron tunnels through a triangular potential barrier. This effect is observed in strong electric fields, at low temperatures, and when the potential barrier values are large.

The Frenkel effect consists of lowering the Coulomb potential of an isolated trap in an electric field (Figure 20d). This effect is observed with a low trap concentration [162]. The Hill–Adachi model of overlapping Coulomb centers has a place at a high trap concentration, so that the trap ionization probability is described by the Pool law: lg(*J*)~*F* (Figure 20e) [163,164]. In the Makram–Ebeid–Lannoo model of multiphonon trap ionization, the electron goes into an excited state and then tunnels into the conduction band (Figure 20f) [165]. This model is implemented with a low trap concentration. In the Nasyrov–Gritsenko model of phonon-assisted tunneling between neighboring traps, an electron from an excited state tunnels to an adjacent trap (Figure 20g). This model is applicable at high trap concentrations [166].

An extensive study of methyl-terminated PECVD OSG films, developed for industrial BEOL applications, was conducted using planar capacitor techniques [139,143]. These studies have led to the conclusion that Poole–Frenkel (PF) emission is predominantly the conduction mechanism in low-*k* dielectrics at low fields [167,168], while Fowler–Nordheim (FN) tunneling conduction is valid at high field ranges [145,169]. Moreover, leakage currents can be used as a direct measure of dielectric structure quality, allowing the examination of the influences of deposition conditions and integration processes. 

Recently, spin-on deposited PMO low-*k* materials with a carbon bridge in their matrix, as well as methyl-terminated spin-on deposited OSG and PECVD OSG, have been studied using capacitors with Mg electrodes and doped Si with a continuous Al contact deposited on the backside of the silicon substrate [101,170,171]. An important feature of these films is that they were cured without using UV and VUV light, and the concentration of adsorbed moisture was carefully reduced through thermal annealing before the measurements. By using these procedures, we aimed to study the low-*k* matrices without the impacts from porogen residue and adsorbed moisture. For three low-*k* dielectric synthesis technologies, an exponential increase in current density was found with an increase in temperature from 300 K to 350 K at fixed high electrical fields. This indicates that charge transport through low-*k* dielectric films cannot be correctly described by the FN mechanism, since this mechanism does not take temperature into account.

It is shown that experimental current–voltage characteristics measured at different temperature (*I*-*V*-*T*) curves for OSG low-*k* dielectrics synthesized by the abovementioned technologies can be described by the Frenkel model with free variation of model parameters. However, the agreement between the theoretical simulation and the experiment is achieved only when using the non-physical value of the pre-exponential parameters *ν* or *N_t_*, as well as the greatly overestimated dielectric constant value *ε*_∞_ (Equation (9)). By using the *ν* = *W*/*h* ≈ 10^14^ s^−1^, when the trap ionization energy *W* is about 1 eV, the *I*-*V*-*T* curves calculated within the Frenkel model describe the experimental ones using the unphysically low trap concentration *N_t_* ≈ 10^2^–10^11^ cm^−3^. And vice versa, by using typical physically justified *N_t_* values under simulation, it is necessary to use unphysically low *ν* values ~10^2^–10^8^ s^−1^. In addition, fitting the theoretical *I-V-T* curves to experimental ones gives values of *ε_∞_* = 6–32, whereas a reasonable value for low-*k* dielectrics hardly exceeds 2 (*ε*_∞_ = *n*^2^ = 1.6). Thus, the Frenkel model can describe the charge transport in the OSG low-*k* dielectric only formally, namely without quantitative agreement of the model’s parameters.

The same statement also turns out to be valid when describing charge transport in OSG low-*k* dielectrics by the Hill–Adachi model of overlapping Coulomb potentials of neighboring traps, as well as the Macram–Ebeid and Lanno model of isolated neutral trap multiphonon ionization to the conduction band [163,164,165]. In the first case, the agreement of the calculated *I*-*V*-*T* with the experimental ones is achieved only when using an unphysically small attempt-to-escape factor value *ν*; in the second case, when using the trap concentration value that lies beyond the model applicability. It was shown that the most consistent model describing the experimental *I*-*V*-*T* curves for OSG low-*k* dielectrics synthesized by three technologies is the phonon-assisted electron tunneling between neighboring neutral traps proposed by Nasyrov and Gritsenko (Figure 19a). The current density in this case is represented by the equation [166]:(9)$$J=\frac{2eN_{t}\sqrt{\pi }\hbar W_{t}}{m^{*}a\sqrt{2kT\left(W_{opt}-W_{t}\right)}}\times \;\times exp\left(-\frac{W_{opt}-W_{t}}{kT}\right)exp\left(-\frac{2a\sqrt{2m^{*}W_{t}}}{\hbar }\right)sinh\left(\frac{eFa}{2kT}\right)$$Here, *W_t_*—thermal trap energy, *W_opt_*—optical trap energy, which is 2 × *W_t_*. It should be noted that the *W_t_* value is unambiguously found in the N-G model from the ln(*J*)-*F* temperature shift, as well as *N_t_* is set by the ln(*J*)-*F* slope.

It is established that for PMO carbon-bridged films, the *W_t_* = 1.6 eV. This value, as well as *W_opt_* = 3.2 eV, coincides with the corresponding energy of the trap responsible for charge transport in SiO_2_ [172]. Since the traps with the specified *W_t_* and *W_opt_* values in SiO_2_ are Si–Si bonds (i.e., oxygen vacancies), one can assume that the traps in PMO carbon-bridged low-*k* dielectrics are Si–Si bonds too. It is noteworthy that the N-G model refinement, by taking into account the space charge through the Poisson equation and charge carrier kinetics through the Shockley–Reed–Hall equations, does not lead to a change in the theoretical *W_t_* and *W_opt_* values. The N-G model refinement only unprincipledly reduces the *m** to 0.4 *m_e_* and increases the *N_t_* to *N* = 3 × 10^21^ cm^−3^.

The charge transport mechanism in the spin-on deposited methyl-terminated OSG low-*k* dielectrics is described by the N-G model with *W_t_* = 1.2 eV and *W_opt_* = 2.4 eV. The trap ionization energy of 1.2 eV indicates that the traps in PECVD methyl-terminated OSG low-*k* dielectrics may be oxygen divacancies [105]. It is hypothesized that the defect with *W_t_* = 1.2 eV is a Si–Si–Si bond (i.e., oxygen divacancies), by analogy with SiO_2_. 

One can see the trap concentration for films deposited by three different technologies is about 10^20^ cm^−3^. Note that for PECVD film, the current below 3.5 MV/cm is at the limit of the device sensitivity threshold (10 pA), and the maximum field for all structures is pre-breakdown. According to the data in Figure 21, the PECVD OSG has the smallest conductivity and the highest breakdown voltage. At the same time, however, the trap concentration for this film is higher than for spin-on deposited methyl-terminated OSG. This discrepancy can be explained by using the N-G model in simulation without the Poisson and Shockley–Reed–Hall equations. The higher leakage currents through carbon-bridged PMO are explained by the highest trap concentration, even despite the greater depth of the trap. 

It can be assumed that ODC centers, including Si–Si or Si–Si–Si bonds, can form in OSG films and influence their conductivity. VUV irradiation of OSG films can rupture Si–CH*_x_* bonds, leading to the formation of Si dangling bonds that can further recombine to form Si–Si or Si–Si–Si defects after structural rearrangement. In contrast, the reduction in conductivity after pure thermal annealing can be attributed to the lack of Si–CH*_x_* bond rupture. However, it is important to note that the processing temperature is constrained by BEOL requirements (<450 °C), which is insufficient to allow the network to relax. This limited thermal budget is particularly important for sol–gel-based films. Sol–gel chemistry relies on stable precursors, with polymerization occurring through hydrolysis and subsequent condensation, forming Si–O–Si or Si–R–Si bridges and methyl groups on the pore wall surface. Therefore, the only possibility of their formation can be related to the collapse of micropores Si dangling bonds during the overcuring (Figure 13) [133], and this process brings the Si dangling bonds to a distance sufficient for their recombination. The negative curvature of the pore wall is an important factor [150,151]. However, this assumption needs additional verification because the detection of ODC centers is not a simple task [94,135]. XPS- and UV-initiated luminescence are widely used in the case of dense oxides, but these methods can meet certain challenges when used for ODC verification in porous OSG films. More detailed information related to these challenges can be found in the Appendix A.

### 4.3. Dielectric Breakdown of Low-k

The analysis of reliability, particularly in assessing the lifetime of integrated dielectrics in ULSI devices, is crucial during their development. The reliability researchers often utilize Weibull [156] or lognormal [173] statistics to understand the lifetime distributions of dielectrics. For a deeper understanding, especially in the evaluation of low-*k* dielectrics integrated with copper in the BEOL structures, additional insights are available in the referenced literature [109].

In this context, we focus on how the lifetime trend relates to applied bias or electric field concerning the types and porosity of OSG-based low-*k* materials. Figure 22 displays the relationship between the characteristic lifetimes of various low-*k* dielectrics plotted against the applied electric field, revealing a clear trend of lower breakdown performance with decreasing *k* value. The degradation of low-*k* properties with a reduction in dielectric constant is conspicuous. This reduction in dielectric constant in films with similar composition is primarily attributed to increased porosity, as per the Clausius–Mossotti equation [6]. The pore size typically varies within a limited range, typically around 1–1.5 nm, leading to an increase in pore wall surface area with increased porosity. Considering this, it is reasonable to assume that early breakdown is primarily linked to contaminations and defects present on the pore wall. This conclusion aligns with previous observations.

However, the escalation of porosity leads to diminished breakdown fields (Figure 19b), a phenomenon elucidated by the augmented presence of cage structures within the film. This exacerbates the strain on the Si–O–Si backbone structure under the influence of an external electric field, accompanied by local field enhancements proximal to the pores. Consequently, the Si–O bond becomes notably susceptible to breakage.

## 5. Conclusions

The main purpose of this paper is to find the correlation between electrical properties (breakdown field and leakage current) and intrinsic defects formed during the fabrication and plasma processing of OSGs. To meet this goal, the second and third sections provide a review of deposition processes and their modification by UV photons and plasma, which are commonly used during technological processing. Both gas-phase (plasma) and liquid-phase (sol–gel technology and spin-on deposition) methods are discussed. Additionally, the methods for generating porosity, types of sacrificial porogens/templates, and UV curing of the deposited films are explored. The differences between PECVD and sol–gel fabricated OSGs are discussed, highlighting the roles of methyl terminal groups and carbon bridging groups. Moreover, the methods for generating porosity, types of sacrificial porogens/templates, and UV curing of the deposited films are explored. The differences between PECVD and sol–gel fabricated OSGs are discussed, highlighting the roles of methyl terminal groups and carbon bridging groups. Additionally, the modifications of materials resulting from the use of various plasma systems and processes employed for micropatterning, which are crucial for practical applications, are presented. For a more detailed analysis of these phenomena, refer to the review papers [25,38], which provide more specific and extensive information.

The third section introduces the modification of materials resulting from the utilization of various plasma systems and processes employed for micropatterning, which are crucial for practical applications. This section explains the features and analysis of VUV-induced modification, detailing the energy and mechanisms involved in bond breakage (quantum chemistry calculations).

The last section includes an analysis of data pertaining to electrical properties. It is demonstrated that the matrix of OSG films containing methyl terminal groups exhibits an optical bandgap and breakdown field similar to that of amorphous SiO₂. This similarity arises from the methyl groups being exclusively situated on the pore wall surface. Moreover, the incorporation of carbon-based bridging groups, in lieu of oxygen bridges, within the silica matrix leads to a reduction in the optical bandgap. This reduction is more pronounced when aromatic groups are employed as bridges compared to alkyl chains. The breakdown field diminishes with increasing porosity. In most cases of porogen-based low-*k* materials, this reduction is typically attributed to a higher concentration of porogen residue in highly porous films. The porogen residue contributes to the formation of a valence band tail, and this effect intensifies after plasma treatment and/or ion bombardment.

Significant efforts have been dedicated to understanding the pivotal role in defining leakage current in various low-*k* materials. The results of different studies clearly demonstrate the significant impact of porosity. A clear increase in leakage current with porosity (pore wall surface) allows us to conclude that the key factor in the degradation of leakage current is related to porogen residue formed during non-optimized UV-assisted thermal curing. The presence of adsorbed moisture also has a strong impact on the leakage current, and its influence depends on the degree of hydrophilicity and, in some cases, can be even stronger than the impact of porogen residue. If the films do not contain adsorbed water molecules and porogen residue on their pore wall surface, the leakage current does not depend on porosity. In this case, the key role in the leakage current mechanism can be attributed to various internal defects (such as Si vacancies, carbon dangling bonds, and oxygen-deficient centers), as described in refs. [88,97]. In most cases, we exclude the formation of ODC centers in low-*k* films due to the very low thermal budget dictated by interconnect technology requirements. The temperatures involved are much lower than those typically needed for matrix relaxation (≥1000 °C) [25], which is a key requirement for ODC formation. Moreover, a significant amount of atomic hydrogen is released during UV curing due to the destruction of CH_3_ groups. Even if silicon dangling bonds are formed by breaking SiCH_3_ bonds, they are immediately saturated by hydrogen, forming stable SiH groups.

Part of the experiments was carried out using specially prepared low-*k* films that did not contain carbon and water residues on their pore wall surfaces [134,148]. In these cases, conductivity becomes independent of porosity and primarily depends on defects in the matrix. The electrical leakage behavior in these cases is well described by the Nasyrov–Gritsenko model [166], which assumes phonon-assisted tunneling between the traps. It has been assumed that in the case of dense dielectrics, these traps are most likely oxygen-deficient centers (ODCs). The formation of ODC centers in low-*k* films has been reported in several papers [94,97,98,135], but it is difficult to confirm because the temperature required for low-*k* fabrication, dictated by the requirements of ULSI interconnect technology, is much lower than the temperature needed for low-*k* matrix relaxation [25]. However, a possible mechanism of their formation in low-*k* dielectrics can be related to the removal of methyl groups from micropores that collapse during overcuring (Figure 13) [133]. In this case, the formed silicon atoms with dangling bonds are located at a short distance due to the negative curvature of micropores. The increased reactivity of chemical groups (hydroxyl groups) in pores with small diameters has already been studied experimentally and discussed [150,151].

In previous research, leakage current has typically been attributed to the Poole–Frenkel mechanism at low electric fields [167,168] and the Fowler–Nordheim mechanism at high electric fields [145,169]. However, our study proposes that the Nasyrov–Gritsenko model [166], which involves phonon-assisted electron tunneling between adjacent neutral traps, offers a more precise explanation for charge transport in OSG low-*k* materials. This model, previously primarily applied to high-*k* dielectrics, appears to better describe the behavior observed in our investigation.

## Figures and Tables

**Figure 1 polymers-16-02230-f001:**
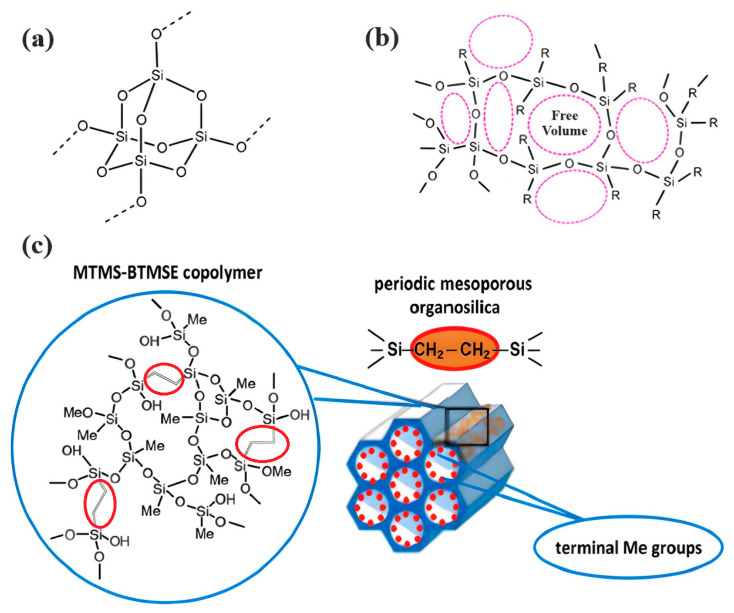
Structure of amorphous SiO_2_ (**a**) and porous methyl-terminated organosilicate glass (OSG) material (**b**), where some oxygen bridging atoms in the SiO_2_ structure are replaced by terminal alkyl groups R. (**c**) Periodic mesoporous organosilica (PMO) with carbon bridges between Si atoms and methyl terminal groups on the pore wall surface. PMO materials are normally synthesized using sol–gel technology.

**Figure 2 polymers-16-02230-f002:**
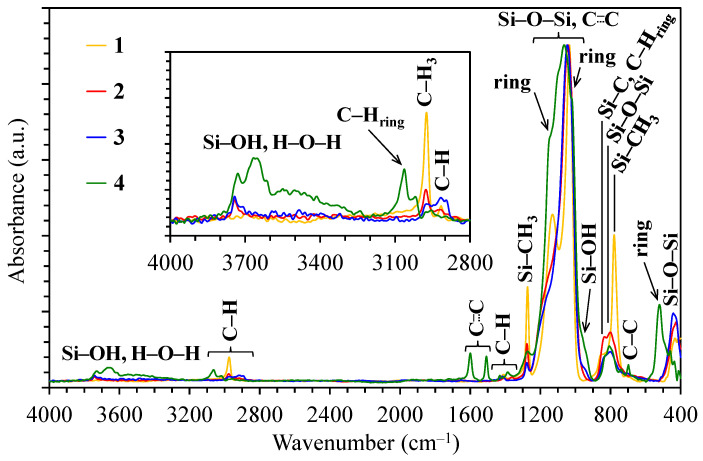
FTIR spectra of organosilicate glass (OSG) films: 1—methylsilsesquioxane (MSSQ), and periodic mesoporous organosilicas (PMOs) with different bridges: 2—methylene, 3—ethylene, 4—1,4-phenylene, 100 mol%, annealed at 430 °C for 30 min in air.

**Figure 3 polymers-16-02230-f003:**
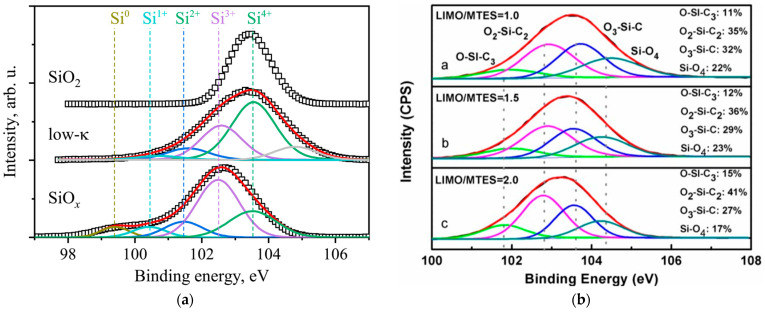
Characteristic X-ray photoelectron spectroscopy (XPS) spectra of the Si 2p peaks for the chemical solution-deposited (CSD) (**a**) and plasma-enhanced chemical vapor-deposited (PECVD) (**b**) methyl-terminated organosilicate glass (OSG) films deposited at different mass flow rate ratios of cinene porogen to triethoxymethylsilane: (a) 1.0; (b) 1.5; (c) 2.0. The presented pictures are redrawn from the data previously reported in our papers [46,47].

**Figure 4 polymers-16-02230-f004:**
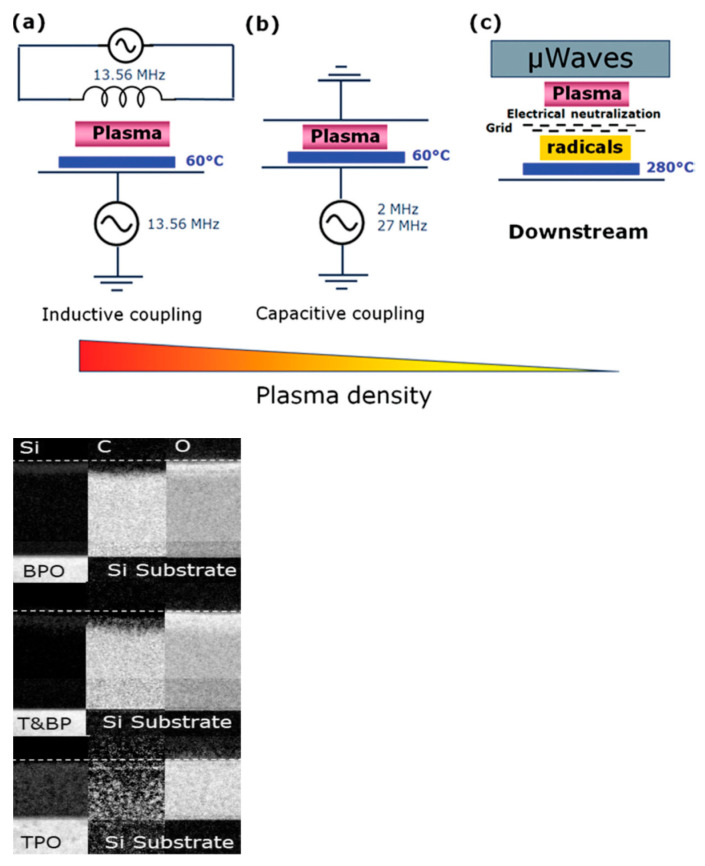
Schematic representation of three different plasma chambers used in microelectronics processing. Inductive coupling plasma (ICP) (**a**) has the highest plasma density and can provide the highest isotropic etch rate, while capacitively coupled plasma (CCP) (**b**) offers a prefect anisotropic profile, but the etch rate is relatively low. For this reason, the reactors combining the ICP and CCP effects are used, and the etch rates and degree of plasma damage can be controlled [55]. Downstream plasma (DSP) (**c**) provides a soft regime and is mostly used for surface cleaning and resist removal when damage-free processing is important. The right picture depicts EFTEM results showing Si, C, and O profiles of low-*k* samples exposed in CCP (BPO), T&BP, and downstream (TPO), and mixed (T&BP) conditions. Reproduced from E. Kunnen, M. R. Baklanov, A. Franquet, D. Shamiryan, T. V. Rakhimova, A. M. Urbanowicz, H. Struyf, W. Boullart; Effect of energetic ions on plasma damage of porous SiCOH low-*k* materials. J. Vac. Sci. Technol. B, 2010; 28 (3): 450–459 [55], with the permission of AVS: Science & Technology of Materials, Interfaces, and Processing.

**Figure 5 polymers-16-02230-f005:**
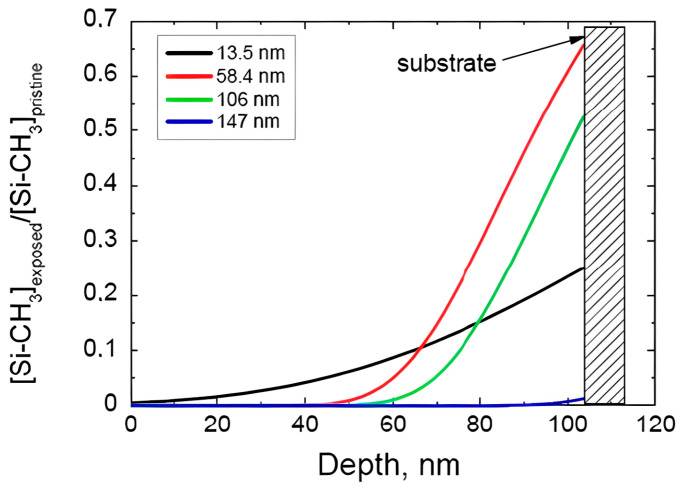
The depth profiles, ranging from the top (0 nm) to the bottom of the film (105 nm), showing the depletion of model Si–CH_3_ bonds in a plasma-enhanced chemical vapor-deposited (PECVD) methyl-terminated organosilicate glass (OSG) film after exposure to 13.5, 58.4, 106, and 147 nm emissions for 7200 s. [Si–CH_3_]_pristine_ refers to the initial SiCH_3_ concentration before exposure to VUV light, while [Si–CH_3_]_exposed_ denotes the SiCH_3_ concentration after VUV exposure. The figure was taken from ref. [83].

**Figure 6 polymers-16-02230-f006:**
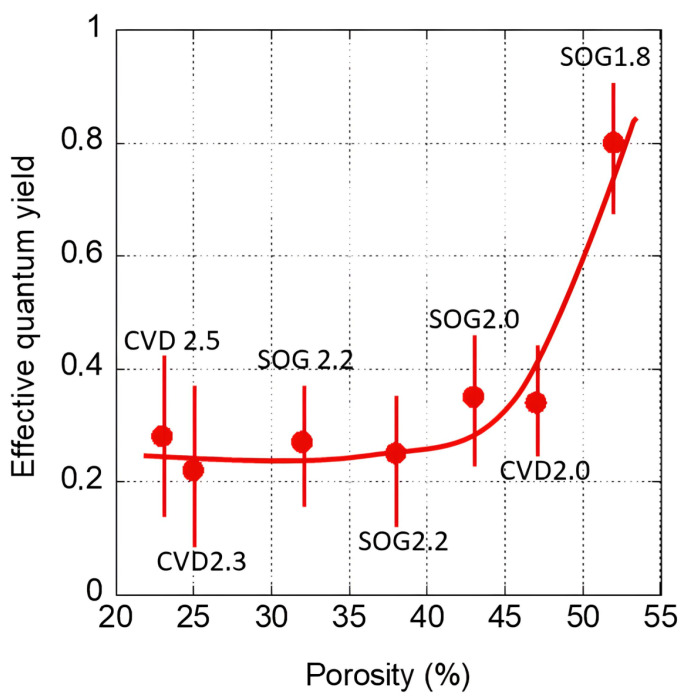
The average effective quantum yield for breaking Si–CH_3_ bonds by VUV photons depending on low-*k* dielectrics porosity. The figure was redrawn based on the data from ref. [83].

**Figure 7 polymers-16-02230-f007:**
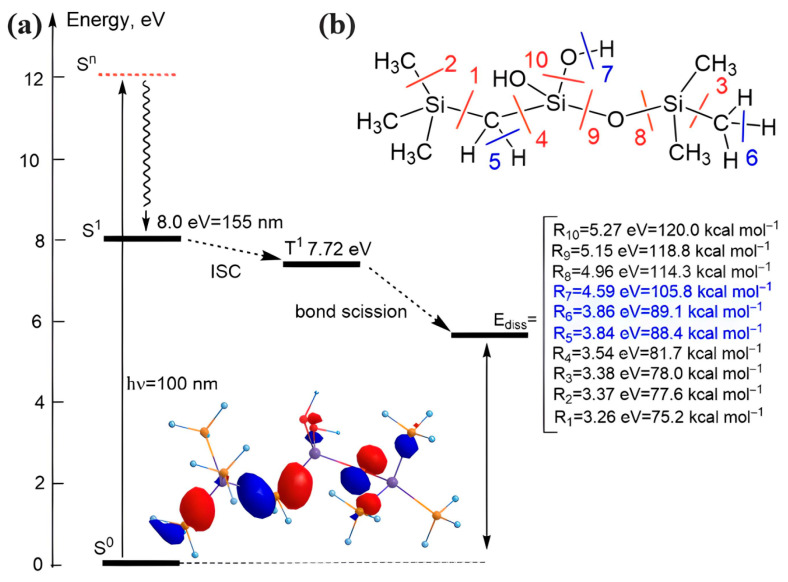
(**a**) Jablonsky diagram depicting electron distribution from the highest occupied molecular orbital (HOMO) in a molecule in a singlet ground state. (**b**) Schema of the possible bond scission in the model periodic mesoporous organosilica (PMO) molecule, with the corresponding dissociation energy calculated as the difference between the free Gibbs energies of the molecule in the ground state and the products of dissociation.

**Figure 8 polymers-16-02230-f008:**
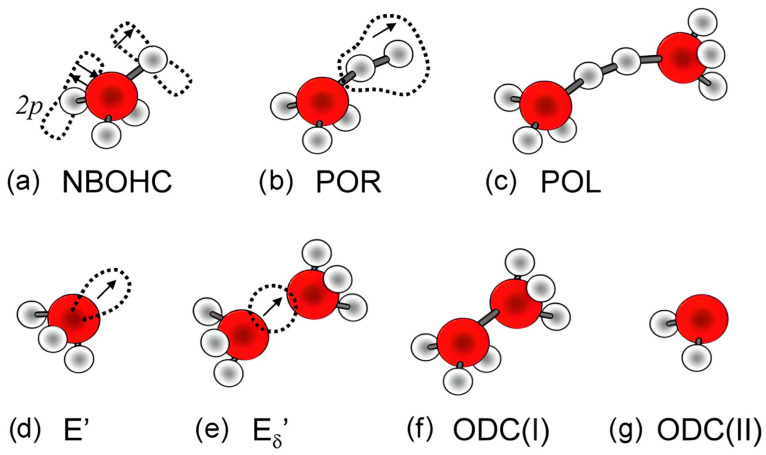
Structural models for SiO_2_ defects: (**a**) non-bridging oxygen hole center, NBOHC; (**b**) peroxy radical, POR; (**c**) peroxy linkage, POL; and defects with a deficit of oxygen: (**d**) E’ and (**e**) E‘_δ_ centers; (**f**) relaxed oxygen vacancy, ODC(I); and (**g**) divalent silicon, ODC(II). Spin states are indicated by the arrows.

**Figure 9 polymers-16-02230-f009:**
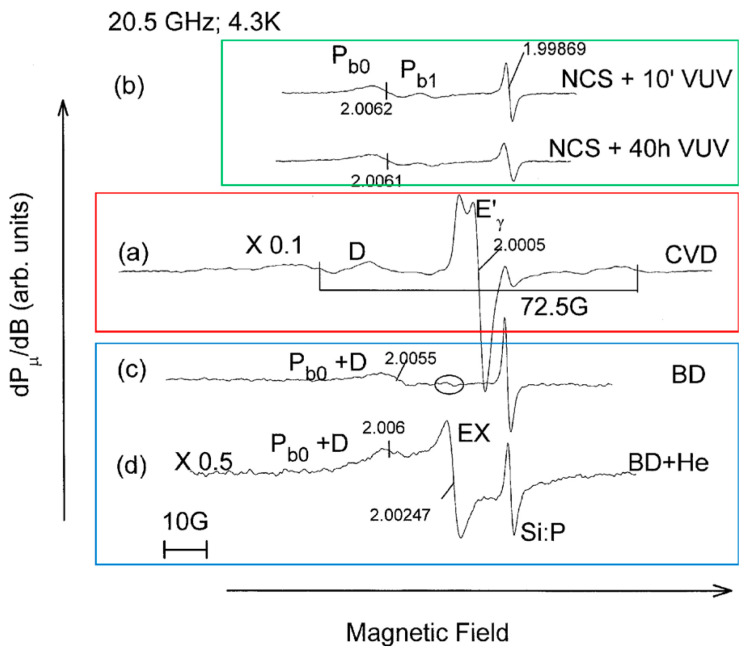
Representative *K*-band electron spin resonance (ESR) spectra measured at 4.3 K on *p*-Si(100) crystal substrates with 200 nm thick layers of chemical vapor-deposited (CVD)-grown *a*-SiO_2_ (CVD, *k* = 4.2) (**a**), nanocrystalline silica (NCS, *k* = 2.3, porosity 30%, pore size ~2 nm) prepared by spin-on coating (**b**), and CVD-grown carbon-doped oxide (BD, *k* = 3.0 and 7% ellipsometric porosimetry (EP)-measured open porosity, pore size ~1.8 nm) without (**c**) and with (**d**) the plasma surface treatment. See ref. [109] for more detail. Reproduced from M. R. Baklanov, V. Jousseaume, T. V. Rakhimova, D. V. Lopaev, Yu. A. Mankelevich, V. V. Afanas’ev, J. L. Shohet, S. W. King, E. T. Ryan; Impact of VUV photons on SiO_2_ and organosilicate low-*k* dielectrics: General behavior, practical applications, and atomic models. Appl. Phys. Rev., 2019; 6 (1): 011301 [25] (Figure 39); and permission for underlying Figure from S. Shamuilia, V. V. Afanas’ev, P. Somers, A. Stesmans, Y.-L. Li, Zs. Tőkei, G. Groeseneken, K. Maex; Internal photoemission of electrons at interfaces of metals with low-*κ* insulators. Appl. Phys. Lett., 2006; 89 (20): 202909 [109] (Figure 3), with the permission of AIP Publishing.

**Figure 10 polymers-16-02230-f010:**
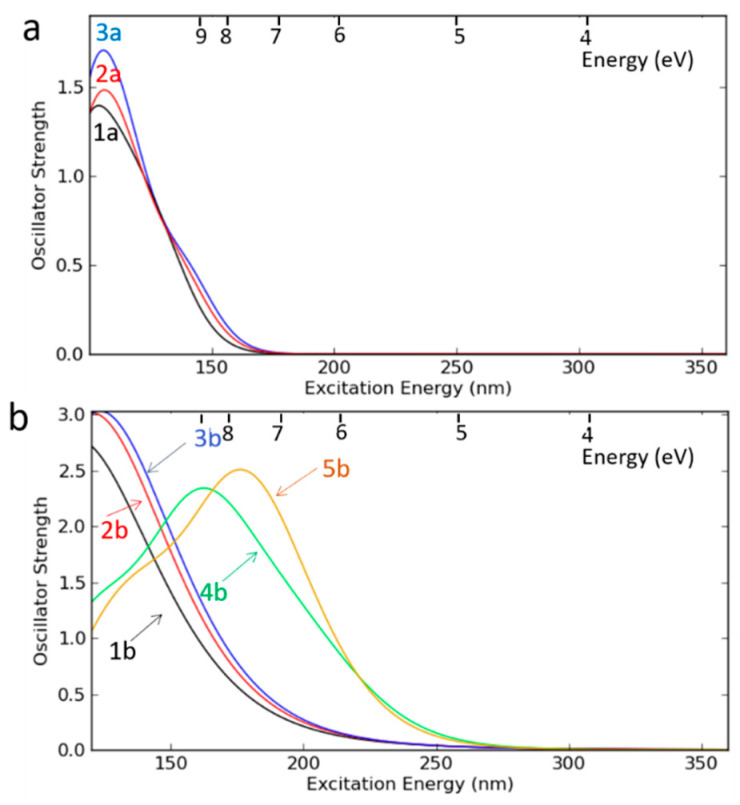
UV absorption spectra of organosilicate glass (OSG) films with various bridging groups shown in Figure 11: 1a—SiO_2_, 2a—OSG with 1 methyl terminal group in the fragment, 3a—OSG with 2 methyl groups, 1b and 2b—one bridging methylene and 6 methyl terminal groups, 3b—ethylene bridge and 6 methyl terminal groups, 4b—1,4-benzene bridge, 5b—hyperconnected 1,3,5-benzene bridge.

**Figure 11 polymers-16-02230-f011:**
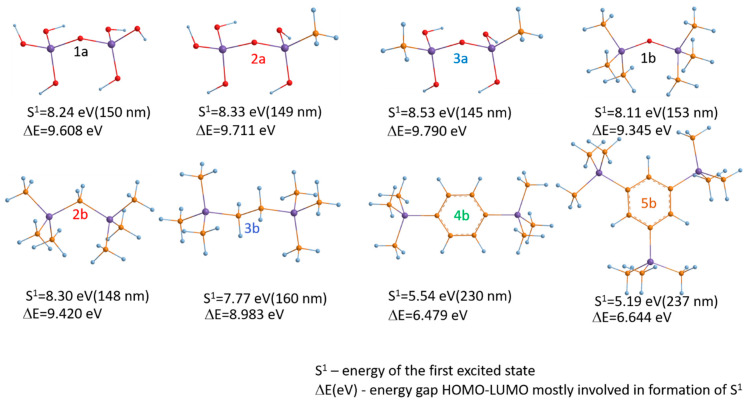
The fragments representing organosilicate glass (OSG) materials with different bridging groups and configurations. The numbers corresponding to the absorption spectra are shown in Figure 10: 1a—SiO_2_, 2a—OSG with 1 methyl terminal group in the fragment, 3a—OSG with 2 methyl groups, 1b and 2b—one bridging methylene and 6 methyl terminal groups, 3b—ethylene bridge and 6 methyl terminal groups, 4b—1,4-benzene bridge, 5b—hyperconnected 1,3,5-benzene bridge. A challenge of such calculations is the selection of an appropriate cluster reflecting the real absorption spectrum of the bulk material. The absorption spectra calculated for the SiO_2_ cluster are in good agreement with the measured spectra, confirming that the calculated spectra are realistic [117,118].

**Figure 12 polymers-16-02230-f012:**
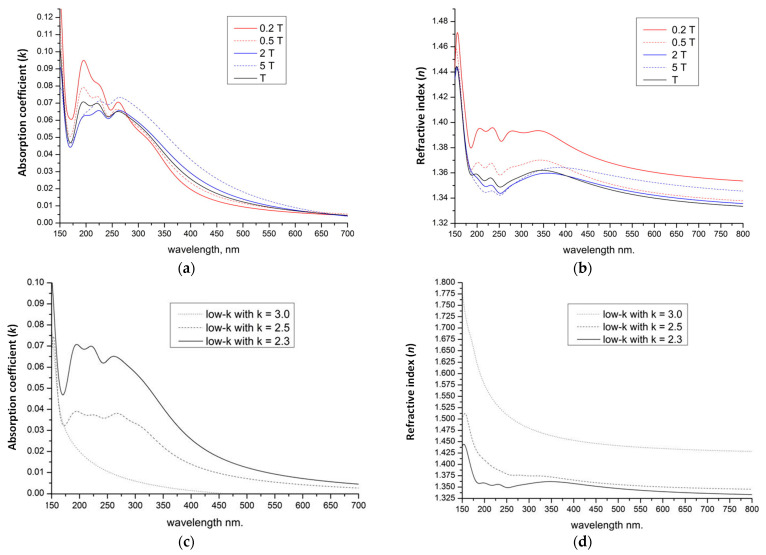
Change in the absorption coefficient and index of refraction of plasma-enhanced chemical vapor-deposited (PECVD) organosilicate glass (OSG) films UV-cured at different times (**a**,**b**) and the films deposited with different porogen concentrations (**c**,**d**). The measured dielectric constant correlates with porosity via the Clausius–Mossotti equation: low dielectric constant corresponds to higher porosity, and therefore, to a higher porogen concentration. T is the optimal curing time used for the fabrication of a standard low-*k* film.

**Figure 13 polymers-16-02230-f013:**
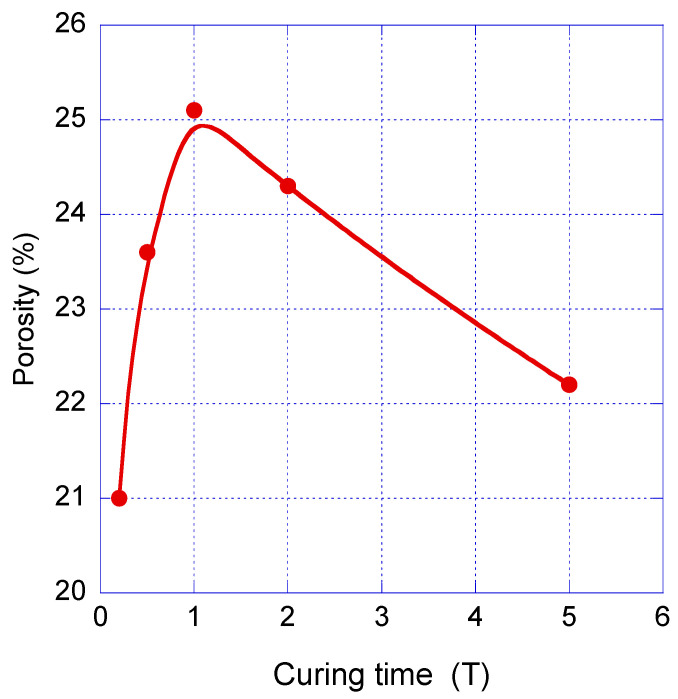
Change in film porosity versus curing time (T). Calculations from the curves presented in Figure 12b using Equation (8). The values of refractive indices at 1.8–2.0 eV are used for the calculation because the extinction coefficient is equal to zero in this region, and Equation (8) is valid.

**Figure 14 polymers-16-02230-f014:**
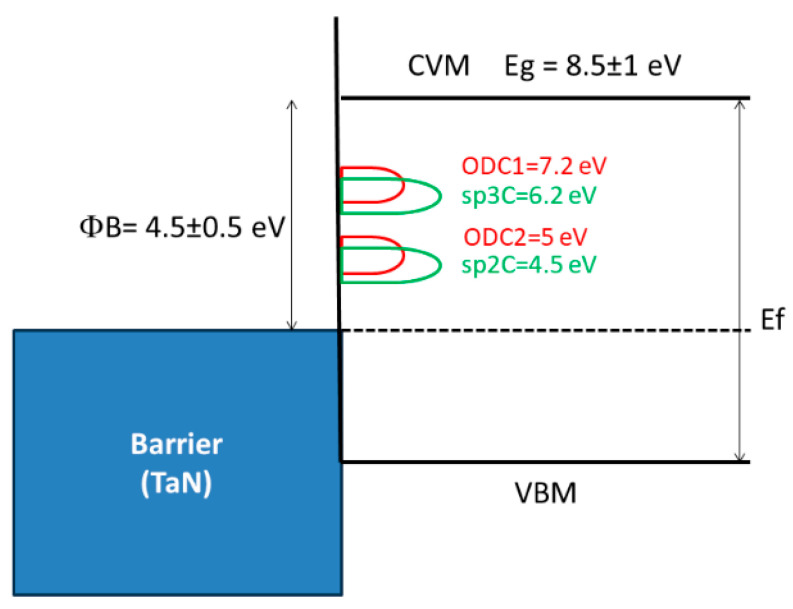
Band alignment for organosilicate glass (OSG) low-*k*/barrier interconnect structure with energy position of defect states related to oxygen-deficient center (ODC) and porogen residues as reported in the papers by King [97] and Marsik [131]. The Schottky barrier between TaN/Ta barrier and low-*k* dielectrics, equal to 4.5 ± 0.5 eV, was measured by using internal photoemission (IPE) measurements by both Shamiulia [109] and Atkin [119].

**Figure 15 polymers-16-02230-f015:**
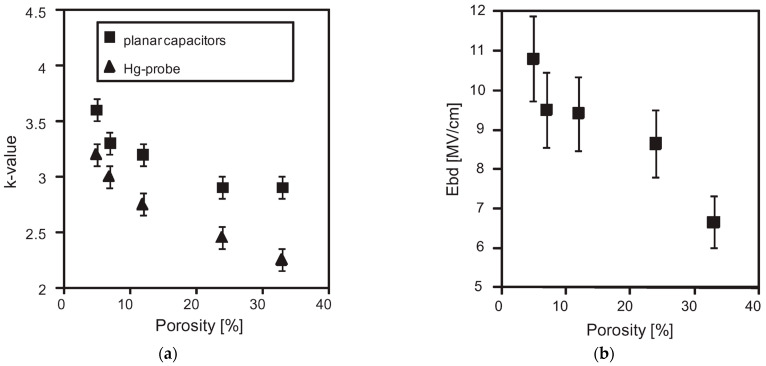
Change in dielectric constant (**a**) and breakdown field (**b**) on porosity of plasma-enhanced chemical vapor-deposited (PECVD) low-*k* films. Reproduced from E. Van Besien, M. Pantouvaki, L. Zhao, D. De Roest, M.R. Baklanov, Z. Tőkei, G. Beyer; Influence of porosity on electrical properties of low-*k* dielectrics. Microelectronic Engineering, 2012, 92: 59–61 [136], with the permission of Elsevier.

**Figure 16 polymers-16-02230-f016:**
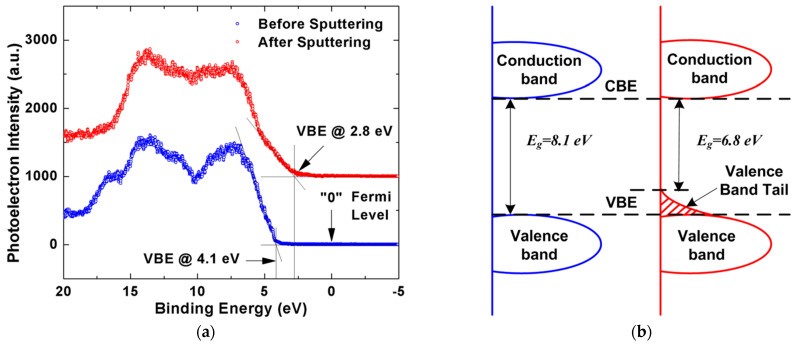
(**a**) Valence band X-ray photoelectron spectroscopy (XPS) spectra of an *a*-SiCOH (*k* = 3.3) film before and after ion sputtering, where the “0” binding energy corresponds to the energy of the Fermi level. (**b**) Schematic representation of the density of states of an *a*-SiCOH (*k* = 3.3) film before and after ion sputtering. Reproduced from X. Guo, H. Zheng, S. W. King, V. V. Afanas’ev, M. R. Baklanov, J.-F. de Marneffe, Y. Nishi, J. L. Shohet; Defect-induced bandgap narrowing in low-*k* dielectrics. Appl. Phys. Lett., 2015; 107 (8): 082903 [135], with the permission of AIP Publishing.

**Figure 17 polymers-16-02230-f017:**
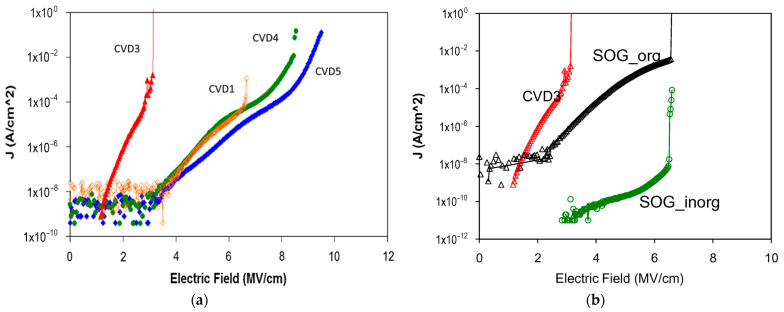
(**a**) The leakage current and breakdown voltage of different types of organosilicate glass (OSG) low-*k* films with *k* values changing from 3.0 (CVD1, CVD4, CVD5) to *k* = 2.3 (CVD3) [144]. (**b**) Comparison of leakage current of CVD3 with organic low-*k* films (Samples 8 and 9 in Table 5) and sol–gel-based SOG films deposited by using a self-assembling approach. Reproduced from M. R. Baklanov, L. Zhao, E. V. Besien, M. Pantouvaki; Effect of porogen residue on electrical characteristics of ultra low-*k* materials. Microelectronic Engineering, 2011, 88: 990–993 [144], with the permission of Elsevier.

**Figure 18 polymers-16-02230-f018:**
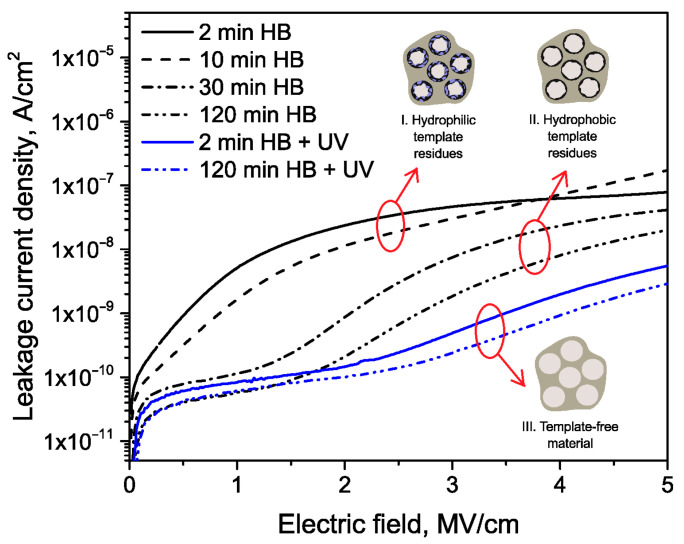
Leakage current density as a function of the applied electric field recorded on metal–insulator–semiconductor (MIS) structures with SOG-2.2 low-*k* films hard-baked (HB)/hard-baked and UV-cured (HB + UV) for different times. Reproduced from M. Krishtab, V. Afanas’ev, A. Stesmans, S. De Gendt; Leakage current induced by surfactant residues in self-assembly based ultralow-*k* dielectric materials. Appl. Phys. Lett., 2017; 111 (3): 032908 [146], with the permission of AIP Publishing.

**Figure 19 polymers-16-02230-f019:**
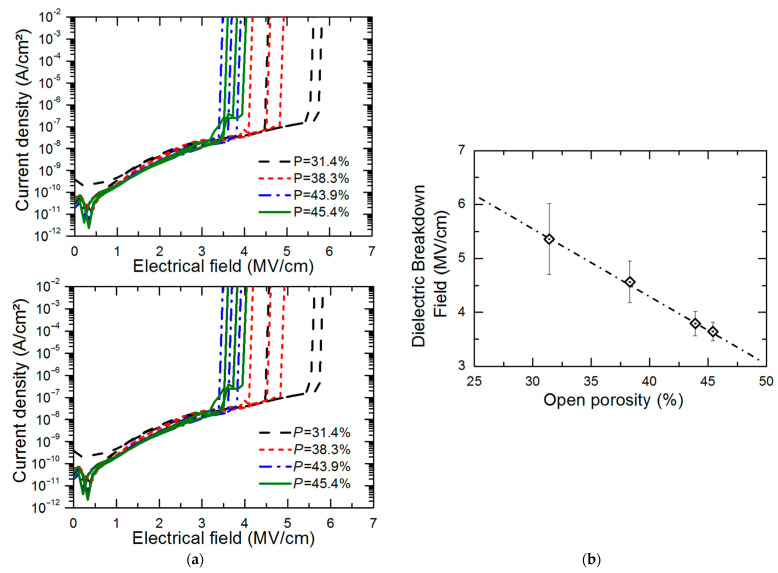
(**a**) Current density as a function of the applied electrical field for porogen residue-free low-*k* films with different levels of porosity, as measured by metal dots. (**b**) Dielectric breakdown field as a function of open porosity at 25 °C. Reproduced from K. Vanstreels, I. Ciofi, Y. Barbarin, M. Baklanov; Influence of porosity on dielectric breakdown of ultralow-*k* dielectrics. J. Vac. Sci. Technol. B, 2013; 31 (5): 050604 [147], with the permission of AVS: Science & Technology of Materials, Interfaces, and Processing.

**Figure 20 polymers-16-02230-f020:**
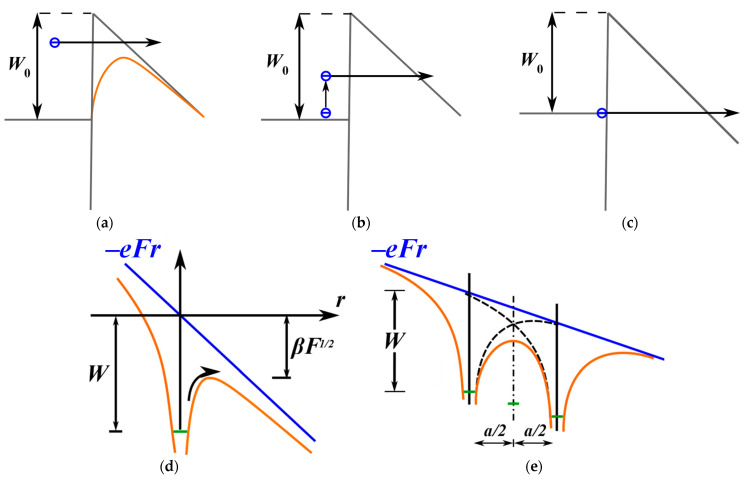
Contact-limited conduction mechanisms: (**a**) Schottky effect, (**b**) thermally assisted tunneling at the contact, (**c**) Fowler–Nordheim effect; bulk-limited conduction mechanisms: (**d**) Frenkel effect, (**e**) Hill–Adachi model, (**f**) Makram–Ebeid and Lanno model, (**g**) Nasyrov–Gritsenko model. Here, *e*—elementary charge, *F*—electric field, *W*_0_—barrier height, *W*—trap ionization energy, *W_t_*—thermal trap ionization energy, *a*—average distance between traps, *E_C_*—conduction band bottom, *E_V_*—valence band top, *β*—is the Frenkel constant, *V_G_*—is the applied voltage.

**Figure 21 polymers-16-02230-f021:**
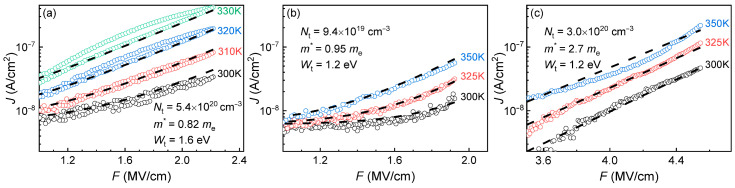
Experimental (characters) and simulations with N-G model (black dash lines) current-voltage characteristics of the (**a**) periodic mesoporous organosilicas (PMO) carbon-bridged low-*k* dielectric [156], (**b**) methyl-terminated spin-on deposited OSG [157], and (**c**) PECVD methyl-terminated organosilicate glass (OSG) low-*k* dielectric [158]. The film thickness is 220 nm and the contact size is 0.5 mm^2^.

**Figure 22 polymers-16-02230-f022:**
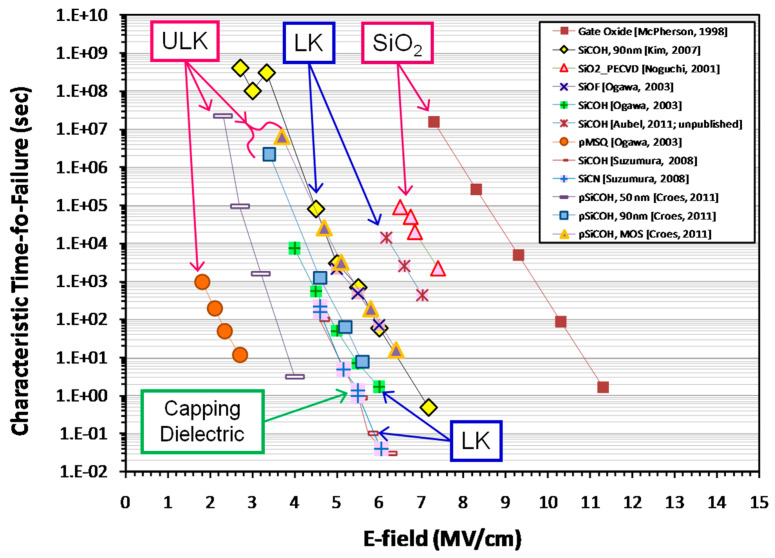
A dielectric breakdown comparison for different low-*k* dielectrics. In this graph, the right two curves reflect SiO_2_ layers fabricated by thermal oxidation of Si and plasma-enhanced chemical vapor-deposited (PECVD) SiO_2_. LK are OSG low-*k* dielectrics with *k* values from 2.5 to 3.0, and ultra-low-*k* (ULK) are low-*k* dielectrics with *k* values from 2.5 to 2.0. The figure is copied from E. T. Ogawa, O. Aubel; Electrical Breakdown. In Advanced Interconnect Dielectrics, 2012; pp. 369–434 [10], with the permission of Wiley & Sons.

**Table 1 polymers-16-02230-t001:** Examples of the matrix and porogen precursors commonly used for chemical vapor deposition (CVD), plasma-enhanced CVD (PECVD), and hot-filament CVD (HFCVD). Examples of their applications, properties of the deposited organosilicate glass (OSG), films, and the corresponding references can be found in ref. [21].

CVD/PECVD/HFCVD Matrix Precursors
Diethoxy-methyl-silane (DEMS)	Tetramethyl-cyclotetrasiloxane (TMCTS)	Deca-methyl-cyclo-pentasiloxane (DMCPS)
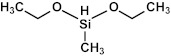	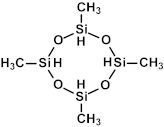	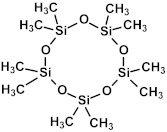
Diethoxy-methyl-oxiranyl-silane	Dimethyl-dioxiranyl-silane	Trimethyl-trivinyl-cyclotrisiloxane (V3D3)
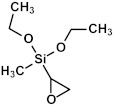	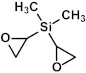	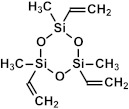
**Porogens**
Norbornadiene (NBD)	Norbornene (NBE)	*a*-terpinene (ATRP)
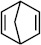	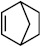	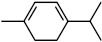
Cyclopentene oxide (CPO)	Cyclohexene oxide (CHO)	Butadiene monoxide (BMO)
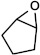	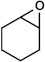	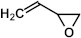

**Table 2 polymers-16-02230-t002:** The most typical examples of matrix precursors and surfactants used for chemical solution deposition (CSD) of organosilica films [1,28,29].

CSD Matrix Precursors
Tetraethoxysilane (TEOS)	Methyltrimethoxysilane (MTMS)	1,2-bis(triethoxysilyl)methane (BTESM)
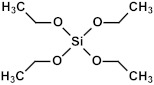	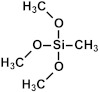	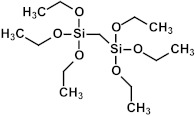
1,2-bis(trimethoxysilyl)ethane (BTMSE)	1,4-bis(triethoxysilyl)benzene (BTESB)	1,3,5-tris(triethoxysilyl)benzene
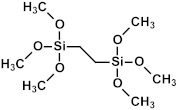	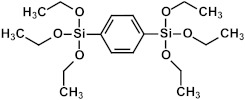	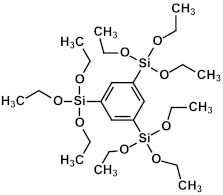
**Surfactants**
Nonionic:	Polyoxyethylene alkyl ethers (Brij^®^)	Poly(ethylene glycol)-poly(propylene glycol)-poly(ethylene glycol) (Pluronic^®^)
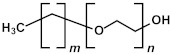 Brij^®^ L4: *m* = 11, *n* = 4;Brij^®^ C2: *m* = 15, *n* = 2;Brij^®^ C10: *m* = 15, *n* = 10;Brij^®^ S10: *m* = 17, *n* = 10	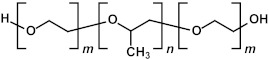 Pluronic^®^ P123: *m* = 20, *n* = 70;Pluronic^®^ F127: *m* = 100, *n* = 65
Ionic:	Alkyltrimethylammonium bromide (C*_n_*TMABr)	Alkyltrimethylammonium chloride (C*_n_*TMACl)
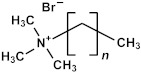 cetyltrimethylammonium bromide (CTAB): *n* = 15;octadecyltrimethylammonium bromide (OTAB): *n* = 17	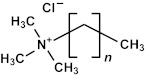 cetyltrimethylammonium chloride (CTAC): *n* = 15;octadecyltrimethylammonium chloride (OTAC): *n* = 17

**Table 3 polymers-16-02230-t003:** Optical bands that are most important for our analysis of intrinsic defects in SiO_2_ and the observed absorption bands in organosilicate glass (OSG) low-*k* films (eV).

Defects	E′	NBOHC	ODC(I)	ODC(II)	sp3 Carbon	sp2 Carbon
Griscom [132]	5.85	4.8	7.6	5.0	-	-
Skuja [99]	5.61	4.7	7.5	5.3	-	-
Marsik [131]	-	-	-	-	6.2	4.5
King [97]	-	-	7.2	5.0	2–6	

E′—oxygen vacancies, NBOHC—non-bridging oxygen hole centers, ODC—oxygen-deficient centers.

**Table 4 polymers-16-02230-t004:** Ion sputtering-related bandgap narrowing in plasma-enhanced chemical vapor-deposited (PECVD) methyl-terminated organosilicate glass (OSG) films with different porosity and different *k* values.

Sample Number	Porosity (%)	*k* Value	Bandgap before Ion Sputtering (eV)	Bandgap after Ion Sputtering (eV)
1	0	3.3	8.1	6.8
2	25	2.5	8.0	6.2
3	34	2.2	8.3	6.1

**Table 5 polymers-16-02230-t005:** Different low-*k* materials used in this research and their characteristics. Curing with a 172 nm VUV light generates much more amorphous carbon-like porogen residue in comparison with broadband UV light with a wavelength (WL) > 200 nm.

Sample Number	Label	Porosity (%)	*K* Value	Curing WL (nm)
1	CVD1	24	2.5	>200
2	CVD2	24	2.5	172
3	CVD2 *	24	2.5	>200
4	CVD3	33	2.3	172
5	CVD3 *	33	2.3	>200
6	CVD4	±5	3.0	No
7	CVD5	n/a	3.2	No
8	SOG_org	30	2.2	No
9	SOG_inorg	35	2.3	No

* Refer to ref. [144].

## Data Availability

The data presented in this study are available upon request from the corresponding authors.

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
