# Peer review of "Comprehensive Review on the Impact of Chemical Composition, Plasma Treatment, and Vacuum Ultraviolet (VUV) Irradiation on the Electrical Properties of Organosilicate Films"

_polymers, 2024, doi:10.3390/polym16152230_

Round 1
Reviewer 1 Report
Comments and Suggestions for Authors
This article looks okay for the review on this field, I have only 3 comments below.
1. Authors should clearly indicate the meaning of [Si-CH3]exposed and [Si-CH3]pristine in figure 5 as there is no definition of those terms in paper. Also define meaning of “depth” in figure 5
2. Line 373 : expression “low-resistivity” should be probably “low-k resistivity”
3. Line 346: expression “low-film” should be “low-k film”
Comments on the Quality of English LanguageEnglish quality does not have any problem.
Author Response
Dear Sir,
Thank you very much for your time and review.
Regarding your 3 comments, we fixed 1 and 3, but we do not change “low-resistivity” (2) because here we are talking about low resistivity metal that is integrated with low-k dielectric.
Comments 1: Authors should clearly indicate the meaning of [Si-CH3]exposed and [Si-CH3]pristine in figure 5 as there is no definition of those terms in paper. Also define meaning of “depth” in figure 5 Response 1: Corrected.
Comments 2: Line 373 : expression “low-resistivity” should be probably “low-k resistivity”
Response 2: NO, we are talking about metal.
Comments 3: Line 346: expression “low-film” should be “low-k film”
Response 3: Corrected.
Sincerely,
Mikhail Baklanov
Alexey Vishnevskiy

Reviewer 2 Report
Comments and Suggestions for Authors
Dear authors,
Understanding the design and fabrication of OSG films is of paramount importance to realize their electrical properties. In this context, the present review article attempts to bridge the preparation method, modification strategies, reaction conditions and the resultant electrical properties. The authors are quite successful in encompassing diverse information in a concrete manner. The listed comments may be considered to further improve the quality of the article.
Minor comments
Introduction
(1) Avoid mentioning the ‘sections’ as ‘chapters’, which may not be the inappropriate terms for a review article.
(2) The previously published review article related to this topic may be overviewed, followed by highlighting the relevance of the present ones.
Section 2.0
(1) The properties of OSG films obtained under different preparation conditions should be specifically highlighted. At present, the general information about the preparation methods is presented without any collective information on the OSG films.
(2) Some information related to the surface roughness of OSG films may be amended.
(3) The textural properties of OSG films obtained under different preparation conditions may be tabulated for more understanding and included in supplementary materials.
(4) The formation mechanisms may be dealt with by reasonable chemical reactions.
Section 3.0
(1) The merit of different etchants used may be outlined.

Author Response
Dear Sir,
Thank you very much for your time and the detailed review.
We implemented the requested changes whenever feasible. However, I would like to provide further clarification on why we conducted this review, despite many aspects having been covered in our previous papers (for instance, in our review paper in Applied Physics Reviews):
- Low dielectric constant materials for microelectronics. Journal of Applied Physics 2003, 93, 8793-8841, doi:10.1063/1.1567460
- Plasma processing of low-k dielectrics. Journal of Applied Physics 2013, 113, 041101, doi: 10.1063/1.4765297
- Impact of VUV photons on SiO2 and organosilicate low-k dielectrics: General behavior, practical applications, and atomic models. Applied Physics Reviews 2019, 6, 011301, doi:10.1063/1.5054304.
We were invited to contribute this paper to the Special Issue "Polymer-SiO2 Composites II," which focuses on a distinct area of research compared to low-k in BEOL. Therefore, prior to delving into electrical properties, we opted to succinctly summarize general information previously discussed in our earlier publications.
Regarding the section on electrical properties, we have described a new concept. Past studies typically attribute leakage current to the Poole–Frenkel mechanism at low fields and the Fowler–Nordheim mechanism at high fields. However, our study indicates that the mechanism of phonon-assisted electron tunneling between neighboring neutral traps describes the charge transport in OSG low-k more accurately. This theory was previously mainly applied to high-k dielectrics.
Furthermore, it is evident that the quality of low-k materials significantly influences leakage current behavior. Materials with residual porogen or adsorbed water on pore walls exhibit current conductivity directly linked to pore surface area. In contrast, "clean" materials, such as those developed by Urbanowicz (Ref. 148 in the manuscript), show leakage current independence from porosity, highlighting the importance of considering internal defects such as oxygen deficient centers (ODC) or similar. Typically, the BEOL thermal budget prevents ODC formation [3], but our study suggests that micropore collapse could create conditions conducive to their formation, though this hypothesis requires further verification.
Comments 1:
Introduction
(1) Avoid mentioning the ‘sections’ as ‘chapters’, which may not be the inappropriate terms for a review article.
(2) The previously published review article related to this topic may be overviewed, followed by highlighting the relevance of the present ones.
Response 1: Fixed, done.
Comments 2:
Section 2.0
(1) The properties of OSG films obtained under different preparation conditions should be specifically highlighted. At present, the general information about the preparation methods is presented without any collective information on the OSG films.
(2) Some information related to the surface roughness of OSG films may be amended.
(3) The textural properties of OSG films obtained under different preparation conditions may be tabulated for more understanding and included in supplementary materials.
(4) The formation mechanisms may be dealt with by reasonable chemical reactions.
Response 2: References [1–3] have already addressed points (1) and (4) extensively. We think that we don’t need to do it here.
As for points (2) and (3), our data on the surface roughness of OSG samples are limited. The observed surface roughness is comparable to that reported for PECVD materials. However, a comprehensive review of various materials would require more extensive analysis and search. Perhaps this is something we can consider for future investigations.
Comments 3:
Section 3.0
(1) The merit of different etchants used may be outlined.
Response 3: This information was discussed in detail in Ref. [2]. Probably we don’t need to do it here.
Sincerely,
Mikhail Baklanov
Alexey Vishnevskiy

Reviewer 3 Report
Comments and Suggestions for Authors
The present review focuses on the impact of the plasma treatment, chemical composition and UV radiation on the electrical properties of the organo-silicate films. I have several concerns related to this study:
1) many review works in the same area has already been published and, more importantly, lead author published recently review on this topic (https://doi.org/10.1063/1.5054304). I would like to ask for novelty related to already published review. I saw that many parts are overlapping .
2) figures and table should be justified where results are taken from.
Overall, authors should clearly stated what novelty present review provides in comparison with previous work.
Comments on the Quality of English Language
English is ok only some slip of pen.
Author Response
Dear Sir,
Thank you very much for your time and the detailed review.
Comments 1:
1) many review works in the same area has already been published and, more importantly, lead author published recently review on this topic (https://doi.org/10.1063/1.5054304). I would like to ask for novelty related to already published review. I saw that many parts are overlapping .
2) figures and table should be justified where results are taken from.
Overall, authors should clearly stated what novelty present review provides in comparison with previous work.
Response 1:
Yes, indeed, many things are overlapping with our previous publications related to material science of low-k dielectrics and published, for instance, in:
- Low dielectric constant materials for microelectronics. Journal of Applied Physics 2003, 93, 8793-8841, doi:10.1063/1.1567460
- Plasma processing of low-k dielectrics. Journal of Applied Physics 2013, 113, 041101, doi: 10.1063/1.4765297
- Impact of VUV photons on SiO2 and organosilicate low-k dielectrics: General behavior, practical applications, and atomic models. Applied Physics Reviews 2019, 6, 011301, doi:10.1063/1.5054304.
The reason for the overlapping is as follows. We were invited to contribute this paper to the Special Issue "Polymer-SiO2 Composites II," which focuses on a distinct area of research compared to low-k in BEOL. Therefore, prior to delving into the analysis of electrical properties, which was the main goal of this paper, we opted to succinctly summarize general information previously discussed in our earlier publications.
Regarding the section on electrical properties, we have described a new concept. Past studies typically attribute leakage current to the Poole–Frenkel mechanism at low fields and the Fowler–Nordheim mechanism at high fields. However, our study indicates that the mechanism of phonon-assisted electron tunneling between neighboring neutral traps describes the charge transport in OSG low-k more accurately. This theory was previously mainly applied to high-k dielectrics.
Furthermore, it is evident that the quality of low-k materials significantly influences leakage current behavior. Materials with residual porogen or adsorbed water on pore walls exhibit current conductivity directly linked to pore surface area. In contrast, "clean" materials, such as those developed by Urbanowicz (Ref. 148 in the manuscript), show leakage current independence from porosity, highlighting the importance of considering internal defects such as oxygen deficient centers (ODC) or similar. Typically, the BEOL thermal budget prevents ODC formation [3], but our study suggests that micropore collapse could create conditions conducive to their formation, though this hypothesis requires further verification.
Sincerely,
Mikhail Baklanov
Alexey Vishnevskiy

Round 2
Reviewer 3 Report
Comments and Suggestions for Authors
Thank you for the explanation. I have no objection to this work. Technically it is fine and sounds well. Overall, the final decision should be done by the editor who did invite authors to contribute to the journal.